
# Advanced source apportionment of carbonaceous aerosols by coupling offline AMS and radiocarbon size segregated measurements over a nearly two-year period.

Athanasia Vlachou[1], Kaspar R. Daellenbach[1], Carlo Bozzetti[1], Benjamin Chazeau[2], Gary A. Salazar[3], Soenke Szidat[3], Jean-Luc Jaffrezo[4], Christoph Hueglin[5], Urs Baltensperger[1], Imad El Haddad[1] and André S.H. Prévôt[1]

[1]Department of General Energy Research, Paul Scherrer Institute, Villigen PSI, CH-5232, Switzerland

[2]Aix Marseille Universite, CNRS, LCE UMR 7376, 13331 Marseille, France

[3]Department of Chemistry and Biochemistry & Oeschger Centre for Climate Change Research, University of Bern, 3012 Bern, Switzerland

[4]Université Grenoble Alpes, CNRS, IRD, G-INP, IGE, 38000 Grenoble, France

[5] Swiss Federal Laboratories for Materials Science and Technology, 8600 Dübendorf, Switzerland.

*Correspondence to*: André S. H. Prévôt (andre.prevot@psi.ch) and Imad El Haddad (imad.el-haddad@psi.ch)

**Abstract.** Carbonaceous aerosols are related to adverse human health effects. Therefore, identification of their sources and analysis of their chemical composition is important. The offline AMS technique offers quantitative separation of organic aerosol (OA) factors that can be related to major OA sources either primary or secondary. While primary OA can be more clearly separated into sources, secondary (SOA) source apportionment is more challenging because different sources - anthropogenic or natural, fossil or non-fossil - can yield similar highly oxygenated mass spectra. Radiocarbon measurements provide unequivocal separation between fossil and non-fossil sources of carbon. Here we coupled these two offline methods and analysed the OA and organic carbon (OC) of different size fractions (particulate matter below 10 and 2.5µm – $PM_{10}$ and $PM_{2.5}$, respectively) from the Alpine valley of Magadino (Switzerland) during the years 2013 and 2014 (219 samples). The combination of the techniques gave further insights into the characteristics of secondary OC (SOC) which was rather based on the type of SOC precursor and not on the volatility or the oxidation state of OC, as typically considered. Out of the primary sources separated in this study, biomass burning OC was the dominant one in winter with average concentrations of 5.36±2.64 µg m$^{-3}$ for $PM_{10}$ and 3.83±1.81 µg m$^{-3}$ for $PM_{2.5}$, indicating that wood combustion particles were predominantly generated in the fine mode. The additional information from the size segregated measurements revealed a primary sulphur containing factor, mainly fossil, detected in the coarse size fraction and related to non-exhaust traffic emissions with average yearly $PM_{10}$ ($PM_{2.5}$) concentration of 0.20±0.24 µg m$^{-3}$ (0.05±0.04 µg m$^{-3}$). A primary biological OC was also detected in the coarse mode peaking in spring and summer with yearly average concentrations for $PM_{10}$ ($PM_{2.5}$) 0.79±0.31 µg m$^{-3}$ (0.24±0.20 µg m$^{-3}$). The secondary OC was separated into two oxygenated, non-fossil OC factors which were identified based on their seasonal variability (i.e. summer and winter OOC) and a third fossil OOC factor which correlated with fossil OC mainly peaking in winter and spring with $PM_{10}$ ($PM_{2.5}$) contributing on average 13%±7% (10%±9%) to the total OC. The winter OOC was connected to anthropogenic sources, with $PM_{10}$ ($PM_{2.5}$) contributing on average 13%±13% (6%±6%) to the total OC. The summer OOC, stemming from oxidation of biogenic emissions was more pronounced in the fine mode with $PM_{10}$ ($PM_{2.5}$) contributing on average 43%±12% (75%±44%) to the total OC. In total the non-fossil OC far dominated the fossil OC throughout all seasons, by contributing on average 75%±24% to the total OC. The results also suggested that during the cold period the prevailing source was residential biomass burning while during the warm period primary biological sources and secondary organic





aerosol from the oxidation of biogenic emissions became important. However, SOC was also formed by aged fossil fuel combustion emissions not only in summer but also during the rest of the year.

## 1. Introduction

The field deployment of the time of flight aerosol mass spectrometer (HR-ToF-AMS, Canagaratna et al., 2007) has advanced our understanding of aerosol chemistry and dynamics. The HR-ToF-AMS provides quantitative mass spectra of the non-refractory particle component, including, but not limited to, organic aerosol (OA), ammonium sulfate and nitrate, by combining the flash vaporization of particle species and the electron ionization of the resulting gases. The application of positive matrix factorization (PMF, Paatero, 1997) techniques has demonstrated that the collected OA mass spectra contain sufficient information to quantitatively distinguish aerosol sources. However, the cost and intensive maintenance requirements of this instrument significantly hinder its systematic, long-term deployment as part of a dense network and most applications are limited to few weeks of measurements (Jimenez et al., 2009, El Haddad et al., 2013, Crippa et al., 2013). This information is critical for model validation and policy directives. The Aerodyne aerosol chemical speciation monitors (ACSM, Ng et al., 2011, Fröhlich et al., 2013, Sosedova et al., in prep) were developed as a low-cost, low-maintenance alternative to the AMS; however their reduced chemical resolution can limit the factor separation achievable by source apportionment.

The recent utilization of the AMS for the offline analysis of ambient filter samples (Daellenbach et al., 2016) has significantly broadened the spatial and temporal scales accessible to high resolution AMS measurements (Daellenbach et al., 2017, Bozzetti et al., 2017). In addition, the technique enables measurement of aerosol composition outside the normal size transmission window of the AMS (the standard AMS can measure up to only 1 µm, or ~2.5 µm with a newly developed aerodynamic lens (Williams et al., 2013, Elser et al., 2016a). This capability has been used to quantify the contributions of primary biological organic aerosol to OA in $PM_{10}$ filters (Bozzetti et al., 2016). Finally, the offline AMS technique allows a retrospective reaction to critical air quality events. For example, one of the applications of this approach had been to examine a severe haze event in China which affected a total area of ~1.3 million $km^2$ and ~800 million people (Huang et al., 2014).

A major limitation of the technique is the resolution of low water solubility fractions, as the recoveries of some of them are not accessible. Despite this, source apportionment results obtained using this technique are in good agreement with online AMS or ACSM measurements. PMF analysis of offline AMS data has yielded factors related with primary emissions from traffic, biomass burning and coal burning and secondary organic aerosols (SOA) differentiated according to their different seasonal contributions. Still, the identification of SOA precursors using the AMS has proven challenging, due to the evolution of different precursors towards chemically similar species and the extensive fragmentation by the electron ionization used in the AMS.

The analysis of radiocarbon ($^{14}C$) is a powerful technique providing an unequivocal distinction between non-fossil (e.g. biomass burning and biogenic emissions) and fossil (e.g. traffic exhaust emissions and coal burning) sources (Szidat et al., 2009). Many studies (Zotter et al., 2014a and 2014b, Zhang et al., 2012, 2016, 2017, Bonvalot et al., 2016, Dusek et al., 2017) measured the $^{14}C$ content of total carbon (TC), which comprises the elemental carbon (EC) originating from combustion sources and the organic carbon (OC) associated with OA. The determination of the $^{14}C$ content in EC and OC separately is challenging and therefore not often attempted for extended datasets, although such measurements are very important especially for the examination of the fossil fraction of OC where EC is actually dominant.

The coupling of the offline AMS/PMF with radiocarbon analysis provides further insights into the sources of organic aerosols and in particular those related to SOA precursors. Such combination has been already attempted (Minguillón et al., 2011, Zotter et al., 2014a, Huang et al., 2014, Beekmann et al., 2015 Ulevicius et al., 2016), however the focus has rather been on high OA concentration episodes, while little is known about the yearly cycle of the most important SOA precursors and the size resolution of the different fossil and non-fossil OA fractions.





Here, we present offline AMS measurements of a total of 219 samples, 154 of which are $PM_{10}$ samples representative of the years 2013 and 2014 and 65 $PM_{2.5}$ concurrent with $PM_{10}$ samples for the year 2014 (January to September). $^{14}C$ analysis was also performed on a subset of 33 $PM_{10}$ samples, covering the year 2014. The size segregated samples offered better insights into the mechanism by which the different fractions enter the atmosphere, while the coupling of offline AMS/PMF and $^{14}C$ analysis provided a more profound understanding of the SOA fossil and non-fossil precursors on a yearly basis.

## 2. Methods

### 2.1 Site and sampling collection

Magadino is located in an Alpine valley in the Southern part of Switzerland, south of the Alps (Figure S1). The station (46° 9´ 37´´ N, 8° 56´ 2´´ E, 204m ASL) belongs to the Swiss national air pollution monitoring network NABEL and is classified as a rural background site. It is located relatively far from busy roads or residential areas and surrounded by agricultural fields and forests. It is ca. 1.4 km away from Cadenazzo train station, ca. 8 km from the lake "Lago Maggiore" and ca. 7 km from the small Locarno airport.

The filter samples under examination are 24 h integrated $PM_{10}$ (from 04/01/2013 to 28/09/2014, with a four-day interval) and $PM_{2.5}$ (from 03/01/2014 to 28/09/2014, with a four-day interval). PM was sampled and collected on 14 cm quartz fibre filters, using a high volume sampler (500 l min$^{-1}$). After the sampling, filter samples and field blanks were wrapped in lint-free paper and stored at -20°C.

### 2.2 Offline-AMS method

The offline-AMS method is thoroughly described by Daellenbach et al. (2016). Briefly, 4 punches of 16 mm diameter from each filter sample are extracted in 15 ml of ultrapure water (18.2 MΩ cm at 25 °C with total organic carbon < 3 ppb), followed by insertion in an ultra-sonic bath for 20 minutes at 30°C. The water extracted samples are then filtered through a 0.45 µm nylon membrane syringe and inserted to an Apex Q nebulizer (Elemental Scientific Inc., Omaha, NE, USA) operating at 60°C. The resulting aerosols generated in Ar (≥ 99.998% Vol., Carbagas, 3073, Gümligen, Switzerland) were dried by a Nafion dryer and subsequently injected and analysed by the HR-ToF-AMS.

To correct for the interference of $NH_4NO_3$ on the $CO_2^+$ signal as described in Pieber et al. (2016), several dilutions of $NH_4NO_3$ in ultrapure water were measured regularly as well. The $CO_2^+$ signal was then calculated as:

$$CO_{2,real} = CO_{2,meas} - \left(\frac{CO_{2,meas}}{NO_{3,meas}}\right)_{NH_4NO_3,pure} * NO_{3,meas} \qquad (1)$$

Where $CO_{2,real}$ represents the corrected $CO_2^+$ signal, $CO_{2,meas}$ and $NO_{3,meas}$ are signals from the samples measured and the correction factor $\left(\frac{CO_{2,meas}}{NO_{3,meas}}\right)_{NH_4NO_3,pure}$ was determined during the campaign by measuring aqueous $NH_4NO_3$.

### 2.3 $^{14}C$ analysis

Based on the instrumentation setup described in Agrios et al. (2015) and on the method described in Zotter et al. (2014b), radiocarbon analysis of TC and EC was conducted on a set of 33 filters. The $^{14}C$ content of blank filters was measured for TC only, as there was no EC found on these filters. All the $^{14}C$ results are given in fractions of modern carbon ($f_M$) representing the $^{14}C/^{12}C$ ratios of each sample relative to the respective $^{14}C/^{12}C$ ratio of the reference year 1950 (Stuiver and Polach, 1977).

### 2.3.1 $^{14}C$ measurements of TC

For the determination of the $^{14}C$ content of TC, a Sunset OC/EC analyser (Model 4L, Sunset Laboratory, USA) equipped with a non-dispersive infrared (NDIR) detector was first used in order to combust each filter punch (1.5 cm$^2$) under pure $O_2$ (99.9995%) at 760 °C for 400 s. The generated $CO_2$ was then captured online by a zeolite trap within a gas inlet system (GIS) and then injected in the accelerator mass spectrometer (*AMS**) MIni





radioCArbon Dating System (MICADAS) at the Laboratory for the Analysis of Radiocarbon with AMS (LARA), University of Bern, Switzerland (Szidat et al., 2014) for [14]C measurement.

The $f_M$ of TC underwent a blank correction following an isotopic mass balance approach:

$$f_{M_{b,cor}} = \frac{mC_{sample} * f_{M,sample} - mC_b * f_{M,b}}{mC_{sample} - mC_b} \qquad (2)$$

where $f_{M_{b,cor}}$ is the blank corrected $f_M$, $mC_{sample}$ and $mC_b$ are the carbon mass in sample and blank, respectively, and $f_{M,sample}$ and $f_{M,b}$ are the $f_M$ measured for sample and blank, respectively. Error propagation was applied for the determination of the $f_{M_{b,cor}}$ uncertainty. The $f_{M,b}$ was 0.61±0.10 and the concentration of the blank 1.1±0.2 µg C m$^{-3}$.

### 2.3.2 [14]C measurements of EC

For the EC isolation of the samples, each filter punch (1.5 cm$^2$) was analysed by the Sunset EC/OC analyser with the use of the Swiss_4S protocol developed by Zhang et al. (2012). According to the protocol, the heating is conducted in four different steps under different gas conditions: step one under pure $O_2$ at 375 °C for 150 s, step two under pure $O_2$ at 475 °C for 180 s, step three under He (>99.999 %) at 450 °C for 180 s followed by an increase of the temperature up to 650 °C for another 180 s and step four under pure $O_2$ at 760 °C for 150 s. Each

filter sample was previously water extracted and dried, in order to minimise the positive artefact induced by the OC by removing the water soluble OC (WSOC), which is known to produce charring (Piazzalunga et al., 2011a, Zhang et al., 2012). By this method, the water insoluble OC (WINSOC) was removed during the first three steps of the Swiss_4S protocol. In the fourth step, EC was combusted and then trapped in the GIS and measured by the *AMS*∗ MICADAS, as described above.

This protocol was preferred over the protocols commonly used in thermo-optical methods (EUSAAR2 or NIOSH) because it optimises the separation of the two fractions OC and EC by minimising i) the positive artefact of charring produced by WSOC during the first three steps and ii) the premature losses, during the removal of the WINSOC in the third step, of the less refractory part of EC which may preferentially originate from non-fossil sources such as biomass burning.

Following a similar principle to Zotter et al. (2014b), both charring and EC yield, which is the part of EC that remained on the filter after step three and before step four in the Swiss_4S protocol, were quantified and corrected for with the help of the laser mounted on the Sunset analyser. The laser transmittance is monitored continuously during the heating process. Charring in step three was quantified as:

$$Charring_{S_3} = \frac{\max ATN_{S_3} - initial\ ATN_{S_2}}{initial\ ATN_{S_1}} \qquad (3)$$

where ATN refers to the laser attenuation.

EC yield in step three was quantified as:

$$ECyield_{S_3} = \frac{initial\ ATN_{S_3}}{\max ATN_{S_3}} * \frac{initial\ ATN_{S_2}}{\max ATN_{S_1}} \qquad (4)$$

The average charred OC was found to be 4±2% and the recovered EC for all samples was on average 71±7%.

As there is a linear relationship between the fraction of modern carbon for EC ($f_{M_{EC}}$) and the EC yield (Zhang et
al., 2012), the slope can be used to extrapolate $f_{M_{EC}}$ to 100% EC yield. According to Zotter et al. (2014), a slope of 0.35±0.11 was considered to correct all $f_{M_{EC}}$ to 100% of EC yield, such that:

$$f_{M_{EC,total}} = slope * \left(1 - ECyield_{S_3}\right) + f_{M_{EC}} \qquad (5)$$





### 2.3.3 Calculation of [14]C content of OC

The fraction of modern carbon of OC ($f_{M_{OC}}$) was calculated following a mass balance approach:

$$f_{M_{OC}} = \frac{TC * f_{M_{TC}} - EC * f_{M_{EC}}}{TC - EC} \tag{6}$$

where TC and EC are the concentrations of total and elemental carbon, respectively, and $f_{M_{TC}}$ and $f_{M_{EC}}$ are the
fractions of modern carbon of TC and EC, respectively. The uncertainty of $f_{M_{OC}}$ was calculated by propagating
the error of each component of Equation (6).

### 2.3.4 Nuclear bomb peak correction

The expected $f_M$ coming from fossil samples should be equal to zero due to the complete decay of [14]C until now,
whereas the $f_M$ from non-fossil samples is expected to be unity. However, due to the extensive nuclear bomb
testing during the late 1950s and early 1960s, the radiocarbon amount in the atmosphere increased dramatically
because of the high neutron flux during the explosions. Therefore the measured $f_M$ of non-fossil samples may
exhibit values greater than one (Levin et al., 2010a). To correct for this effect, the $f_M$ is normalised to a reference
non-fossil fraction ($f_{NF,ref}$) which represents the amount of [14]C currently in the atmosphere compared to 1950,
before the nuclear bomb tests. As EC comes from either biomass burning or fossil sources, the non-fossil
fraction of EC ($f_{NF,EC}$) equals the $f_M$ coming from biomass burning ($f_{M,bb}$). The latter was estimated by a tree
growth model (Mohn et al., 2008) and was equal to 1.101. The non-fossil fraction of OC ($f_{NF,OC}$) is calculated as:

$$f_{NF,OC} = p_{bio} * f_{M,bio} + p_{bb} * f_{M,bb} \tag{7}$$

where $f_{M,bio}$ (=1.023) is the fraction of modern carbon of biogenic sources and was estimated from [14]$CO_2$
measurements in Schauinsland (Levin et al., 2010a). The fractions of biogenic sources ($p_{bio}$) and biomass
burning ($p_{bb}$) to the total non-fossil sources were set to 0.5 since both sources are important in Magadino during
the year (biomass burning in winter, biogenic sources in summer).

### 2.4 Additional measurements

Organic and elemental carbon fractions were determined by a Sunset EC/OC analyser with the use of the
EUSAAR-2 thermal-optical transmittance protocol (Cavalli et al., 2010). Water soluble organic carbon (WSOC)
was measured by a total organic carbon (TOC) analyser (Jaffrezo et al., 2005) with the use of catalytic oxidation
of water extracted filter samples and detection of the resulting $CO_2$ with an NDIR. The concentrations of major
ionic species ($K^+$, $Na^+$, $Mg^{2+}$, $Ca^{2+}$, $NH_4^+$, $Cl^-$, $NO_3^-$ and $SO_4^{2-}$) as well as methane sulfonic acid (MSA) were
determined by ion chromatography (Jaffrezo et al., 1998). Anhydrous sugars (levoglucosan, mannosan,
galactosan) were analysed by an ion chromatograph (Dionex ICS1000) using high-performance anion exchange
chromatography (HPAEC) with pulsed amperometric detection. Cellulose was analysed by performing
enzymatic conversion of cellulose to D-glucose (Kunit and Puxbaum, 1996) and D-glucose was determined by
HPAEC.

## 3. Source apportionment

### 3.1 Method

The obtained organic mass spectra from the offline-AMS measurements were analysed by positive matrix
factorization (PMF) (Paatero and Tapper, 1994; Ulbrich et al., 2009). PMF attempts to solve the bilinear matrix
equation:

$$X_{ij} = \sum_k G_{i,k} F_{k,j} + E_{i,j} \tag{8}$$

by following the weighted least squares approach. In the case of aerosol mass spectrometry, $i$ represent the time
index, $j$ the fragment and $k$ the factor number. If $X_{ij}$ is the matrix of the organic mass spectral data and $s_{i,j}$ the
corresponding error matrix, $G_{i,k}$ the matrix of the factor time-series, $F_{k,j}$ the matrix of the factor profiles and $E_{i,j}$
the model residual matrix, then PMF determines $G_{i,k}$ and $F_{k,j}$ such that the ratio of the Frobenius norm of $E_{i,j}$
over $s_{i,j}$ is minimised. The allowed $G_{i,k}$ and $F_{k,j}$ are always non-negative. The input error matrix $s_{i,j}$ includes the





measurement uncertainty (ion counting statistics and ion-to-ion signal variability at the detector) (Allan et al., 2003) as well as the blank variability. Fragments with a signal-to-noise ratio (SNR) below 0.2 were removed and the ones with SNR lower than 2 were down-weighted by a factor of 3, as recommended by Paatero and Hopke, (2003). Both input data and error matrices were scaled to the calculated water soluble OM ($WSOM_i$)

concentration:

$$WSOM_i = \frac{OM}{OC} * WSOC_i \tag{9}$$

where $\frac{OM}{OC}$ is determined from the AMS measurements and $WSOC_i$ is the water soluble OC measured by the TOC analyser.

The Source Finder toolkit (SoFi v.4.9, Canonaco et al., 2013) for IGOR Pro software package (Wavemetrics,

Inc., Portland, OR, USA) was used to run the PMF algorithm. The PMF was solved by the Multilinear Engine 2 (ME-2, Paatero, 1999), which allows the constraining of the $F_{k,j}$ elements to vary within a certain range defined by the scalar α ($0 \le α \le 1$), such that the modelled $F'_{k,j}$ equals:

$$F'_{k,j} = F_{k,j} \pm α * F_{k,j} \tag{10}$$

Here we constrained only the hydrocarbon-like factor (HOA) from high resolution mass spectra analysed by

Crippa et al., 2013b.

### 3.2 Sensitivity analysis

To understand the variability of our dataset we explored 4-10 factor solutions and retained the 7 factor solution as the best representation of the data. The exploration of the PMF solutions is thoroughly described in section S.1.

We have assessed the accuracy of PMF results by bootstrapping the input data (Davison and Hinkley, 1997). New input data and error matrices were created by randomly resampling the time-series from the original input matrix (223 samples in total: 219 + 4 re-measurements from the PM$_{10}$ samples), with replacement; i.e. any sample from the whole population can be resampled more than once. Each sample measurement included on average blocks of 12 mass spectral repetitions, therefore resampling was performed on the blocks. Out of the

223 original samples, some of them were several times represented, while some others not at all. Overall, the resampled data made up on average 64±2% of the total original data per bootstrap run. We performed 180 bootstrap runs, with each of the generated matrices being perturbed by varying the $X_{ij}$ element within twice the corresponding error matrix $s_{i,j}$. Within the resampling operation, the α value used to set the HOA constraining strength was varied between 0 and 1 with an increment of 0.1, to assess the sensitivity of the results on α value.

To select the physically plausible solutions we applied two criteria:

1) We accepted solutions where the average absolute concentrations of all factors in PM$_{2.5}$ did not statistically significantly exceed their concentrations in PM$_{10}$. For this we performed a paired *t*-test with a significance level of 0.01 (Figure S2 and Table S1).
2) We excluded outlier solutions identified by examining the correlation of factor time series from

bootstrap runs with their respective factor time series from the average of all bootstrap runs. The rejected solutions included factors that did not correlate with the corresponding average factor time series meaning that one of the factors was not separated (Figure S3 in the case of PBOA).

In total 24 bootstrap runs were retained after the application of the aforementioned criteria.

### 3.3 Recoveries

In order to rescale the WSOC concentration of a factor $k$ to its total concentration OC$_k$, we used factor recoveries ($R_k$) as proposed by Daellenbach et al., 2016. First, the $WSOM_k$ was calculated as:

$$WSOM_k = f_{k,WSOM} * WSOC_{measured} * \left(\frac{OM}{OC}\right)_{bulk} \tag{11}$$



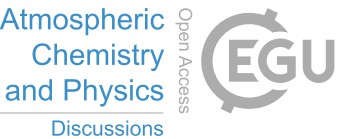

$$\text{where } f_{k,WSOM} = \frac{WSOM_{k,measured}}{\sum_k WSOM_{k,measured}} \tag{12}$$

The $WSOM_k$ was converted to $WSOC_k$ to fit the measured OC concentrations (determined by the Sunset EC/OC analyser). The $WSOC_k$ was determined as:

$$WSOC_k = \frac{f_{k,WSOM} * WSOC_{measured} * \left(\frac{OM}{OC}\right)_{bulk}}{\left(\frac{OM}{OC}\right)_k} \tag{13}$$

Finally, the recoveries were applied following equation (14):

$$OC_{i,k} = \frac{WSOC_{i,k}}{R_k} \tag{14}$$

To assess the recoveries and their uncertainties, we evaluated the sum of $OC_{i,k}$ against the measured OC ($OC_{i,measured}$) by fitting Equation (15). The starting values for the $R_k$ fitting were based on Bozzetti et al. (2016) (for $R_{PBOA}$) and Daellenbach et al. (2016) except $R_{SCOA}$ which was randomly varied between 0 and 1 (increment: $10^{-4}$). While $R_{HOA}$ and $R_{SCOA}$ were constrained, $R_{PBOA}$, $R_{BBOA}$, $R_{WOOA}$, $R_{FOOA}$ and $R_{SOOA}$ were determined by a non-negative multilinear fit. The multilinear fit was chosen to be non-negative because a negative $R_k$ would mean negative concentration of $WSOC_k$ or $OC_k$. The fit was performed for 100 times for each of the retained bootstrap solutions.

$$OC_{i,measured} = \sum_k \frac{WSOC_{i,k}}{R_k} \tag{15}$$

Each fit was initiated by perturbing the $OC_{i,k}$ and the $WSOC_{i,k}$ concentrations within their uncertainties assuming a normal distribution of errors, to assess the influence of measurement precision on $R_k$. Additionally, we introduced a constant 5% accuracy bias corresponding to the OC and WSOC measurement accuracy.

To select the environmentally meaningful solutions we applied the following criteria:

1) To retain the recoveries that achieved the OC mass closure, we estimated the OC residuals and discarded solutions where OC residuals were statistically different than 0 within 1 standard deviation for each size fraction individually and for winter and summer individually.
2) We also examined the dependence between the WSOC residuals and each factor $WSOC_{i,k}$ ($t$-test, $\alpha = 0.001$). Overall, 55% of the solutions were retained.
3) The physically plausible range of the recoveries is [0, 1]. However, the mathematically possible range can exceed the upper limit. $R_k$ larger than 1 would mean that $WSOC_k$ is larger than $OC_k$ and is, therefore, non-physical. For that reason, out of the accepted solutions that survived the previous two criteria, the retained $R_k$ combinations were weighted according to their physical interpretability. More specifically, fitting results with $R_k$ larger than 1 were down weighted according to the measurement uncertainties of WSOC and OC (see S.2, Fig S4).

## 4. Results and discussion

### 4.1 PM$_{10}$ composition

PM$_{10}$ in Magadino has been characterized by high carbonaceous concentrations during winter (Gianini et al., 2012a, Zotter et al., 2014b). This is clearly illustrated in Fig. 1 where an overview of the PM$_{10}$ composition is presented in Fig. 1a with Fig.1b and 1c summarizing the concentrations and relative contributions of each component to the total PM$_{10}$ averaged per season. The peaks of OM and EC during winter (daily averages up to 26 µg m$^{-3}$ and 5.9 µg m$^{-3}$, respectively) are indications of the increased wood burning activity. Other Alpine sites close to Magadino, such as Roveredo and San Vittore in Switzerland, have also exhibited high OM concentrations due to residential wood burning (Szidat et al., 2007 for PM$_{10}$ in Roveredo, Lanz et al., 2010 for PM$_1$ in Roveredo and Zotter et al., 2014b for PM$_{10}$ in San Vittore and Roveredo). The organic contribution dominated the inorganic fraction not only in winter, but also throughout both years (Fig.1c). Note that the EC



concentrations are much lower in spring compared to winter (Fig.1b). The main inorganic aerosols contributing to the total PM, occurring as $(NH_4)_2SO_4$ and $NH_4NO_3$ are $NO_3^-$, $SO_4^{2-}$ and $NH_4^+$. $NO_3^-$ represented the second major component of $PM_{10}$, exhibiting a seasonal cycle with higher concentrations during winter (2.9 µg m$^{-3}$). The notable discrepancy of $NO_3^-$ concentrations between the first (2013) and second (2014) winter could be

explained by the lower temperatures in January – February 2013 compared to 2014. Conversely, $SO_4^{2-}$ showed a rather stable yearly cycle with slightly higher concentrations in summer (1.9 µg m$^{-3}$) compared to winter (1.3 µg m$^{-3}$), despite a shallower boundary layer height in winter.

### 4.2 $^{14}$C analysis results

So far radiocarbon results have been reported mostly for relatively short periods of time (Bonvallot et al., 2016),

mainly describing high-load events and there are few studies that report yearly cycles (Zotter et al., 2014b, Zhang et al., 2016; Zhang et al., 2017; Dusek et al., 2017). Here, for a subset of 33 $PM_{10}$ filters from the year 2014, we present yearly cycles of $OC_{nf}$, $OC_f$, $EC_{nf}$ and $EC_f$.

Overall the total carbon (TC) concentrations followed a yearly pattern mainly caused by the shallow planetary boundary layer and the enhanced biomass burning activity during winter, with OC reaching on average (±one

standard deviation) 9.36±4.52 µg m$^{-3}$ and EC 2.58±1.48 µg m$^{-3}$ (Fig. 2a). During the rest of the year, TC remained rather stable with much lower concentrations ($OC_{avg}$ = 3.70±1.90 µg m$^{-3}$ and $EC_{avg}$ = 0.80±0.69 µg m$^{-3}$). $^{14}$C results indicate that non-fossil sources prevail over fossil in Magadino. More specifically, we found that in winter on average $f_{NF,OC}$ = 0.89±0.06 and $f_{NF,EC}$ = 0.52±0.10 which is in agreement with the reported fractions by Zotter et al., 2014b ($f_{NF,OC}$ = 0.82±0.07 and $f_{NF,EC}$ = 0.49±0.15 ). Table 1 summarizes the $f_{NF}$ per fraction

season-wise.

$OC_{nf}$ was the dominant part of TC throughout the year with contributions of up to 80% in winter and 71% in summer (Fig., 2b) and averaged concentrations of 8.48±4.23 µg m$^{-3}$ and 2.39±0.56 µg m$^{-3}$ in winter and summer, respectively (Fig. 3b). Such high contributions in winter strongly indicate that biomass burning (BB) from residential heating is the main source of carbonaceous aerosols in that region, as has already been observed

before (Jaffrezo et al., 2005, Favez et al., 2010, Zotter et al., 2014b). The coefficient of determination $R^2$ between $OC_{nf}$ and levoglucosan, a characteristic marker for BB, was 0.92 (Fig. S7a) and the slope ($OC_{nf}$/levoglucosan = 4.82±0.27) lies within the reported range by Zotter et al. (2014b) for Magadino (which was 6.9±2.6).

The concentration of $EC_{nf}$ was significantly higher in winter (average 1.31±0.66 µg m$^{-3}$) compared to the rest of

the year (spring average: 0.35±0.23 µg m$^{-3}$, summer average: 0.21±0.06 µg m$^{-3}$, autumn average: 0.43±0.41 µg m$^{-3}$) (Fig. 3d). $EC_{nf}$ is considered to originate solely from BB, for instance from residential wood burning in winter. This assumption is supported by the very high correlation ($R^2$=0.95) with levoglucosan (Fig. S7b) and the slope ($EC_{nf}$/levoglucosan=0.82±0.03) which is also in agreement with literature (Zotter et al., 2014b, Herich et al., 2014).

The strong correlation between $OC_{nf}$ and $EC_{nf}$, driven mainly by the winter data points, supports the fact that $OC_{nf}$ is mostly from biomass burning in winter (Fig. S6a). In late spring, summer and early autumn, the contribution of $EC_{nf}$ decreased significantly (on average to 0.23±0.07 µg m$^{-3}$). The low correlation of $OC_{nf}$ and $EC_{nf}$ during this period (Fig., S6a), in combination with the increase of the $OC_{nf}$/$EC_{nf}$ ratio in summer (Fig. 3b), suggests that a part of the secondary $OC_{nf}$ originates from non-combustion sources, e.g. biogenic/natural

sources.

In total, the relative contribution of the fossil fraction to the TC was 27%. Excluding winter, $EC_f$ exhibited slightly higher concentrations than $EC_{nf}$ (Fig. 3c and d). The average concentrations of $EC_f$ were 1.26±0.93 µg m$^{-3}$, 0.41±0.35 µg m$^{-3}$, 0.31±0.07 µg m$^{-3}$ and 0.63±0.56 µg m$^{-3}$ for winter, spring, summer and autumn, respectively (Fig. 3c). The increase of $EC_f$ witnessed in winter could be mainly attributed to the shallower

planetary boundary layer (PBL) rather than to an increase in the emissions (Fig. S8a). The sources of $EC_f$ in the coarse ($PM_{10}$ - $PM_{2.5}$) size fraction are typically related to resuspension of abrasion products of vehicle tires or brake wear (Bukowieki et al., 2010, Zhang et al, 2013). The fine part of $EC_f$ is due to fossil fuel burning, here



mostly due to traffic exhaust emissions. It is significantly correlated with NOx (Fig. S8b) and the $EC_f/NOx = 0.020$ ratio lies within the reported slopes (Zotter et al. 2014b and references therein).

The contribution of $OC_f$ to TC decreased during winter (8%) but remained roughly stable throughout the rest of the year (22% in spring, 21% in summer and 19% in autumn, Fig. 2b) with average concentrations 0.87±0.30 μg
m$^{-3}$, 0.96±0.12 μg m$^{-3}$, 0.89±0.14 μg m$^{-3}$, 0.76±0.10 μg m$^{-3}$ for winter, spring, summer and autumn, respectively (Fig. 3a). The low correlation overall observed between $OC_f$ and $EC_f$ (Fig. S6b) may indicate that a fraction of $OC_f$ is not directly emitted but formed as secondary OC from fossil fuel related emissions (e.g. traffic). This is supported by low $OC_f/EC_f$ ratios in winter (on average 0.7±0.3) and much higher values in spring and summer (on average 2.7±1.1) (Fig. 3a). The low ratios are consistent with tunnel measurement studies (Li et al., 2016,
Chirico et al., 2011, El Haddad et al., 2009) and the increase of $OC_f/EC_f$ in spring/summer above these values is an indication of anthropogenic SOA formation.

**4.3 Offline-AMS analysis results**

**Factor interpretation**
In this section, we will interpret the PMF outputs. The factor recoveries for all factors, $R_k$, determined as
described in Section 3.3, are shown in Figure 4. Factor mass spectra are displayed in Figure 5. The contribution of the different factors to OA is presented in Figure 6. In addition, for some cases we will discuss the factor contribution to OC, to check the consistency of our results with previous literature reports. Recoveries values determined and used in this study will also be compared for each factor to previous values. Median values of the recoveries as well as the OM/OC ratios with their interquartile range are compiled in Table 2. The $R_k$ values
were in general consistent with previous reports (Daellenbach et al., 2016 and 2017, Bozzetti et al., 2016). Here we report for the first time the recoveries of each SOA factor individually which were in agreement with the low end of the $R_{OOA}$ (which includes the sum of all oxygenated factors), reported by Daellenbach et al., 2016. The consistency of the recoveries results with not only previous offline AMS/PMF studies but also with online AMS measurements (Xu et al., 2017), points out that this method is rather robust and universal for different datasets.

HOA, typically associated with traffic emissions, was constrained using the reference HOA high resolution profile from Crippa et al., 2013b. The resulting factor profile (Fig. 5) exhibited a low OM/OC (Table 2) and the time series followed the one from NOx (Fig. 6). As the offline AMS technique requires water extracted samples, it is expected that HOA, which mostly contains water-insoluble material, will be poorly represented. This is also shown by the low recovery $R_{HOA,median}$ which was estimated to be 0.11 ($Q_{25} = 0.10$ and $Q_{75} = 0.13$) as reported in
Daellenbach et al. (2016) (Fig. 4). Therefore, the correlation between HOA and NOx was weak (Fig. S9). However, the HOA/NOx ratio was 0.017 for $PM_{10}$ and 0.008 for $PM_{2.5}$ and these values are consistent with already reported ones in literature (Daellenbach et al., 2017, Lanz et al., 2007). In addition, the HOC time series followed a similar yearly cycle as $EC_f$ (Fig. S10a) and the $HOC/OC_f$ ratio was 0.37±0.12 (Fig. S10b), in agreement with Zotter et al. (2014a).

BBOA was identified by its significant contributions of the oxygenated fragments $C_2H_4O_2^+$ (at $m/z$ 60) and $C_3H_5O_2^+$ (at $m/z$ 73), common markers for wood burning formed by fragmentation of anhydrous sugars (Alfarra et al., 2007) (Fig. 5). It was also identified by its distinct seasonal variation which exhibited exclusively high concentrations in winter, reaching up to 20.0±0.7 μg m$^{-3}$ for $PM_{10}$ in December 2013 and 12.3±0.5 μg m$^{-3}$ for $PM_{2.5}$ in January 2014 (Fig. 6). The median value for the OM/OC ratio was 1.77 and the $R_{BBOA}$ was consistent
with the low end of the reported one by Daellenbach et al., 2016 (Table 2). The identification of this factor as BBOA was further confirmed by its remarkable correlation with levoglucosan. Similar to levoglucosan, this factor did not exhibit a significant difference between $PM_{2.5}$ and $PM_{10}$ concentrations (Fig. S5a), suggesting that most of these particles are present in the fine mode, consistent with previous observations (Levin et al., 2010b). The BBOA/levoglucosan ratio was 7.12 for $PM_{10}$ and 5.75 for $PM_{2.5}$, which falls into the range reported by
Daellenbach et al. (2017) and was also consistent with the ratio reported by Bozzetti et al. (2016). The difference of BBOA/levoglucosan for the two size fractions is due to four samples in BBOA $PM_{10}$ with high concentrations. Lastly, BBOC showed a strong correlation with $EC_{nf}$, with a slope of 4.87 (Fig. 7b) which fell within the range of the compiled $EC_{nf}$/BBOC ratios in Ulevicious et al. (2016).



SCOA was identified by its spectral fingerprint which is described by a high contribution of the fragment $CH_3SO_2^+$ (at $m/z$ 79) (Fig. 5) and high OM/OC ratio (Table 2). The $R_{SCOA}$ (Fig. 4, Table 2) showed a much broader distribution than the rest primary OC recoveries, yet more limited towards the strongly water-soluble fractions compared to Daellenbach et al. (2017). SCOA concentrations were higher in the coarse fraction

compared to $PM_{2.5}$ (Fig. 6 and 7c, Fig. S5) and exhibited higher concentrations during autumn and winter compared to summer (Table 3). A similar profile had been linked to marine origin by Crippa et al. (2013b) in Paris, however, Daellenbach et al. (2017) found that SCOA in Switzerland was rather a primary locally emitted source with no marine origin due to its anti-correlation with methane sulfonic acid (MSA). Here we confirm that SCOA did not follow the MSA time-series (Fig. S11) but rather the time-series of NOx. These observations

suggest that this factor is connected to a primary coarse particle episodic source related to traffic.

PBOA exhibited significant contributions from the fragment $C_2H_5O_2^+$ (part of $m/z$ 61) (Fig.5) and was more enhanced in summer and spring (Fig. 6). The $R_{PBOA}$ (Fig. 4, Table 2) met the high end of $R_{PBOA}$ in Bozzetti et al. (2016). PBOA appeared mostly in the coarse mode (Table 3, Fig. S5). The mass spectral features, the seasonality and coarse contribution suggested the biological nature of this factor possibly including plant debris.

Additional support of this interpretation is provided by the correlation of PBOA with cellulose (Fig. 7d), a polymer mostly found in the cell wall of plants. The correlation improved if data only from summer and spring were considered. The outliers here were the late autumn and winter points when BBOA was more important and PBOA could not as easily be separated by the PMF technique.

FOOA was identified as a highly oxidized factor, due to the significant contribution of the fragment $CO_2^+$ (Fig.

5) and the high OM/OC ratio (Table 2) which was consistent with the reported OM/OC ratio by Turpin et al. (2001) for non-urban aerosols. This factor peaked mainly in winter and spring and the $PM_{2.5}$ size fraction exhibited higher concentrations during this period compared to coarse size fraction (Table 3, Fig. 6). The water solubility of FOOA was high (Fig. 4, Table 2), consistent with literature values (Daellenbach et al.2016 and 2017) that refer to the sum of all oxygenated factors, as well as with reported water-soluble fractions for highly

oxidized compounds (Xu et al, 2017). The yearly median concentration for $PM_{10}$ was 0.97 µg m$^{-3}$ (Q25=0.86 µg m$^{-3}$ and Q75=1.09 µg m$^{-3}$) which accounts for approximately 13% of the total OA. Out of all the possible correlations with external markers, FOOC correlated best with $OC_f$ (Fig. 7e) and both followed very similar annual cycles (Fig. S12) with average $FOOC/OC_f$ = 0.97±2.49. This observation along with the increase of $OC_f/EC_f$ as already discussed in section 4.2 could indicate that this factor is linked to secondary organic aerosol

from traffic emissions. Further discussion about FOOC can be found in Section 4.4.

SOOA was mainly identified by the high contribution of the fragment $C_2H_3O^+$ ($m/z$ 43) (Fig. 5) ($fC_2H_3O^+$=0.15) as well as its seasonal behavior (Fig. 6). Like all the oxygenated OA factors, it was highly water soluble (Fig. 4, Table 2). The highest concentrations were witnessed in July with values of 4.43 µgm$^{-3}$ for $PM_{10}$ in 2013 and 4.27 µgm$^{-3}$ for $PM_{2.5}$ in 2014. The bulk of this factor contribution was present in the $PM_{2.5}$ fraction (Table 3, Fig.

S5). The seasonal variability of SOOA followed the daily averages of temperature seasonal variability (Fig. 6). In fact, SOOA exponentially increased with temperature (Fig. 7d). Such behaviour was also observed in Daellenbach et al. (2017), where they connected this factor to the oxidation of terpene emissions and therefore to biogenic SOA formation. The exponential dependence of SOOA with temperature was also similar with the temperature dependence of the biogenic SOA concentrations from a Canadian terpene-rich forest, reported by

Leaitch et al. (2011). A similar factor was identified with an online instrument in Zurich during summer 2011, where the semi-volatile OOA was mainly formed by biogenic sources as the high temperatures favour the biogenic emissions compared to the rest (Canonaco et al., 2015). Finally, the O:C ratio (0.37) fell into the range of the reported O:C ratios measured by chamber generated SOA (Aiken et al., 2008), which was similar to biogenic SOA produced in flow tubes (Heaton et al., 2007).

Named after its seasonal behavior (Daellenbach et al. 2017), the third oxygenated factor, WOOA, exhibited the highest concentrations during winter. WOOA mass spectrum exhibited elevated contributions of the fragment $C_2H_3O^+$ (Fig. 5), but lower compared to SOOA (for WOOA $fC_2H_3O^+$=0.11). It also exhibited a slightly enhanced contribution of the fragment $C_2H_4O_2^+$ which can be an indication that this factor originated from aged biomass burning emissions. Moreover, a similar mass spectral pattern (peaks of fragments $C_3H_3O^+$, $C_3H_5O_2^+$, $C_4H_5O_2^+$





and $C_5H_7O_2^+$ at $m/z$ 55, 73, 85 and 99, respectively) with the one coming from oxygenated products from a wood burning experiment (Bruns et al., 2015) was found. The recovery of this factor manifested high values (Table 2) and consisted mainly of fine mode particles (Fig. S5). WOOA also correlated with $NH_4^+$ (Fig. S13), directly connected to the inorganic secondary ions $NO_3^-$ and $SO_4^{2-}$.

### 4.4 Coupling of offline AMS and $^{14}$C analyses

In this section of the paper we will show the combined results of AMS/PMF and radiocarbon analyses. In the first part the technical aspect of the analysis will be elaborated, by presenting the calculation of the contribution of each factor to the fossil OC. In the second part, a thorough description of each fossil and non-fossil major source will be given. The time-series of each fossil and non-fossil fraction for the whole AMS dataset is illustrated in Fig. 10. Contributions of the primary and secondary OC to the total OC will be also discussed and shown in Fig. 11.

#### 4.4.1 Calculation of fossil and non-fossil fraction per factor

To combine the AMS/PMF with the $^{14}$C results, the identified sources from AMS/PMF were divided into fossil and non-fossil fractions. HOC was fully assigned to fossil sources assuming that the percentage of biofuel content is negligible. BBOC and PBOC were considered totally non-fossil. To explore the fossil and non-fossil nature of the rest of the factors, we performed multilinear regression using equation (16):

$$OC_f - HOC = a*SCOC + b*FOOC + c*SOOC + d*WOOC \qquad (16)$$

where a, b, c and d are the fitting coefficients, weighted by the relative uncertainty of $OC_f$ - HOC. To investigate the stability of the solution, we obtained distributions of the fitting coefficients by performing 100 bootstrap runs where input data were randomly selected (Fig. 8). The median values (and 1$^{st}$ and 3$^{rd}$ quartiles) were: a=0.81 ($Q_{25}$=0.73, $Q_{75}$=0.88), b=0.77 ($Q_{25}$=0.54, $Q_{75}$=0.85), c=0.21 ($Q_{25}$=0.15, $Q_{75}$=0.26) and d=0.23 ($Q_{25}$=0.13, $Q_{75}$=0.39).

We chose to apply the multilinear regression to the fossil fraction because for the non-fossil part, the errors related to fitting coefficients were very high and the dependences of the $OC_{nf}$ on the input factors were not statistically significant (p-values > 0.1).

To calculate the non-fossil part of each factor k ($kOC_{nf}$), we used the following equation:

$$kOC_{nf} = kOC - kOC_f \qquad (17)$$

This analysis suggests that the major fossil primary sources were HOC and SCOC (81%±11% fossil), while FOOC (77%±23% fossil) was the only major fossil secondary source. In terms of the non-fossil sources, the dominating primary sources included BBOC and PBOC, whereas the most important secondary sources were SOOC (79%±11% non-fossil) and WOOC (77%±23% non-fossil).

#### 4.4.2 Contribution of fossil and non-fossil, primary and secondary OC to the total OC

The results point out that 81%±11% (average and one standard deviation) of SCOC was fossil ($SCOC_f$). Taking into account the enhanced contribution of SCOC in the coarse size fraction, its sulphur content and its fossil nature, we assume that this factor is linked to primary anthropogenic sources related to traffic, such as resuspension of road dust (Bukowiecki et al., 2010), resuspension from asphalt concrete (Gehrig et al., 2010) or asphalt mixture abrasion (in bituminous binder, Fullova et al., 2017). The contribution of $SCOC_f$ to the $OC_f$ was more important during autumn and winter (up to 62%, Fig. 9a) in contrast to spring and summer (on average 9%±5%), while on average the contribution to the $OC_f$ was 20%±19%. The concentrations in winter and autumn were similar and on average for $PM_{10}$ ($PM_{2.5}$) 0.22±0.21 µg m$^{-3}$ (0.03±0.03 µg m$^{-3}$) (Fig. 10, Table S2) which accounted for 73% of the total SCOC for this period. However, the contribution of $SCOC_f$ to the total OC for the coarse size fraction was not high (5%±8% on average).

The combined $^{14}$C / AMS analysis supported the initial hypothesis that FOOC was mainly related to an anthropogenic origin and more specifically to the oxidation of fossil fuel combustion emissions (e.g. traffic), as $FOOC_f$ was 77%±23% fossil on average. The average contribution of $FOOC_f$ to the $OC_f$ was 28%±14% (Fig.





9a), larger than $SCOC_f$, while its contribution to the total OC was 10%±5% for the coarse OC and 7%±7% of the fine OC. The yearly cycle exhibited elevated contributions in winter and spring compared to summer and autumn with average values for $PM_{10}$: 0.47±0.22 µg m$^{-3}$, 0.43±0.30 µg m$^{-3}$, 0.39±0.23 µg m$^{-3}$ and 0.29±0.23 µg m$^{-3}$, respectively (Fig. 10, Table S2). In winter and spring most of the mass concentration came from the $PM_{2.5}$ size range in contrast to the other two seasons.

The fossil fractions of SOOC and WOOC were low (21% and 23%, respectively) and could also be attributed to traffic emissions or less likely (due to low emissions) to aged aerosols from residential fossil fuel heating. $SOOC_f$ was important during summer with contributions to the $OC_f$ up to 40% and $WOOC_f$ was more distinctively present during a few days in autumn and winter (up to 35% to the $OC_f$) in contrast to the rest of the year (Fig. 9a).

From the non-fossil sources, apart from $SCOC_{nf}$ and $FOOC_{nf}$, the rest of the factors exhibited a very distinct yearly cycle with BBOC contributing up to 86% to the $OC_{nf}$ in late autumn and winter (Fig. 9b, on yearly average 28%±30%) and with PBOC and $SOOC_{nf}$ becoming more important in late spring, summer and early autumn with contributions up to 82% and 57%, respectively (Fig. 9b).

SOOC was 79% non-fossil which supported the AMS/PMF results: the significance of $SOOC_{nf}$ during summer can be attributed to SOA formation from biogenic emissions. The average contribution of $SOOC_{nf}$ to $OC_{nf}$ was 25%±19% (Fig. 9b). $SOOC_{nf}$ was more pronounced in $PM_{2.5}$ (on average 1.12±0.40 µg m$^{-3}$ in summer and 0.75±0.35 µg m$^{-3}$ in spring, Fig. 10, Table S2). This factor along with PBOC was the main and almost equally important source of OC during spring and summer, with PBOC contributing to OC in the coarse mode (on average 35%±16% from April to August 2014) and $SOOC_{nf}$ in the fine mode (46%±15% from April to August 2014). PBOC made up 30%±18% of the $OC_{nf}$ and the average concentrations of $PBOC_{coarse}$ for 2014 were in summer 1.00±0.23 µg m$^{-3}$ and spring 0.56±0.21 µg m$^{-3}$.

$WOOC_{nf}$ dominated over $WOOC_f$ (77% over 23%). The average yearly contribution to $OC_{nf}$, was low (6%±6%, Fig. 9b), however, $WOOC_{nf,coarse}$ was apparent during the cold period especially in 2013 with concentrations of 0.88±0.74 µgm$^{-3}$ on average for winter (0.28±0.28 µg m$^{-3}$ for autumn) (Fig. 10). In 2014 the concentrations dropped for winter (autumn) with 0.53±0.43 µg m$^{-3}$ (0.15±0.13 µg m$^{-3}$) for $PM_{10}$ and 0.22±0.19 µg m$^{-3}$ (0.21±0.21 µg m$^{-3}$) for $PM_{2.5}$. Based on its yearly cycle (Fig. 10b and d) $WOOC_{nf}$ could be linked to aged OA influenced by wintertime and early spring biomass burning emissions. In other studies (Daellenbach et al., 2017, Bozzetti et al., 2016) this factor was more pronounced, however, in our case in winter most of the $OC_{nf}$ was related to primary biomass burning.

Overall for $PM_{10}$ the non-fossil primary OC contributions were more important during autumn (57%) and winter (75%), whereas in spring and summer the non-fossil secondary OC (SOC) contributions became more pronounced (32% and 40%, respectively) (Fig. 11). The dominance of the SOC during the warm period is likely related to the stronger solar radiation which favours the photo-oxidation of primary biogenic organic aerosols and to the elevated biogenic volatile organic compounds emissions.

## 5 Conclusions

The coupling of offline AMS and $^{14}$C analyses allowed a detailed characterisation of the carbonaceous aerosol in the Alpine valley of Magadino for the years 2013-2014. The seasonal variation along with the two size-segregated measurements ($PM_{10}$ and $PM_{2.5}$) gave insights into the source apportionment, by for example quantifying the resuspension of road dust or asphalt concrete and estimating its contribution to the OC or by identifying SOC based on SOC precursors. More specifically, seven sources including four primary and three secondary ones were identified. The non-fossil primary sources were dominating during autumn and winter, with BBOC exhibiting by far the highest concentrations. During spring and summer again two non-fossil sources PBOC in the coarse fraction and $SOOC_{nf}$ in the fine mode, prevailed over the fossil ones. The size-segregated measurements and $^{14}$C analysis enabled a better understanding of the primary SCOC factor, which was enhanced in the coarse fraction and was mainly fossil suggesting that it may originate from resuspension of road dust or tire - asphalt abrasion. The results also showed that SOC was formed mainly by biogenic sources during summer. However, SOC formed by oxidation of traffic emissions was also apparent during summer



(FOOC$_f$). FOOC$_f$ was also important during winter along with SOC linked to aged biomass burning aerosols (WOOC$_{nf}$).

**Acknowledgements**

This work is funded by the Swiss Federal Office for the Environment (FOEN), OSTLUFT and the Cantons Basel, Graubünden, Ticino, Thurgau and Valais. The LABEX OSUG@2020 (ANR-10-LABX-56) funded analytical instruments at IGE.

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





5    Table 1. Median OC and EC non-fossil fractions per season in PM$_{10}$ with interquartile range.

|  | Autumn | | | Winter | | | Spring | | | Summer | | |
|---|---|---|---|---|---|---|---|---|---|---|---|---|
|  | Q$_{25}$ | Q$_{50}$ | Q$_{75}$ | Q$_{25}$ | Q$_{50}$ | Q$_{75}$ | Q$_{25}$ | Q$_{50}$ | Q$_{75}$ | Q$_{25}$ | Q$_{50}$ | Q$_{75}$ |
| $f_{NF,OC}$ | 0.71 | 0.77 | 0.83 | 0.87 | 0.88 | 0.93 | 0.70 | 0.75 | 0.79 | 0.73 | 0.76 | 0.79 |
| $f_{NF,EC}$ | 0.36 | 0.41 | 0.44 | 0.44 | 0.52 | 0.56 | 0.42 | 0.49 | 0.51 | 0.38 | 0.39 | 0.42 |

Table 2. Variability of OM/OC and factor recoveries.

|  | OM/OC | | | R$_k$ | | |
|---|---|---|---|---|---|---|
|  | Q$_{25}$ | Q$_{50}$ | Q$_{75}$ | Q$_{25}$ | Q$_{50}$ | Q$_{75}$ |
| **HOA** | 1.32 | **1.33** | 1.36 | 0.10 | **0.11** | 0.13 |
| **BBOA** | 1.76 | **1.77** | 1.78 | 0.60 | **0.61** | 0.63 |
| **SCOA** | 2.03 | **2.16** | 2.20 | 0.68 | **0.81** | 0.89 |
| **PBOA** | 1.74 | **1.76** | 1.82 | 0.41 | **0.42** | 0.44 |
| **FOOA** | 2.12 | **2.14** | 2.16 | 0.72 | **0.79** | 0.87 |
| **SOOA** | 1.66 | **1.67** | 1.68 | 0.78 | **0.84** | 0.94 |
| **WOOA** | 1.76 | **1.79** | 1.83 | 0.72 | **0.78** | 0.92 |

10    Table 3. Season-wise median concentrations (in µg m$^{-3}$) of different OA factors per size fraction and their
interquartile range in parentheses. Note that for the two different years the months per season can vary.

| µg m$^{-3}$ | Autumn | | | Winter | | | Spring | | | Summer | | |
|---|---|---|---|---|---|---|---|---|---|---|---|---|
|  | 2013 (Sept, Oct, Nov) | 2014 (Sept) | | 2013 (Jan, Feb, Dec) | 2014 (Jan, Feb) | | 2013 (Mar, Apr, May) | 2014 (Mar, Apr, May) | | 2013 (Jun, Jul, Aug) | 2014 (Jun, Jul, Aug) | |
|  | PM10 | PM10 | PM2.5 | PM10 | PM10 | PM2.5 | PM10 | PM10 | PM2.5 | PM10 | PM10 | PM2.5 |
| **HOA** | 0.23±0.20 | 0.46±0.20 | 0.44±0.19 | 1.38±1.35 | 0.45±0.36 | 0.66±0.26 | 0.67±0.55 | 0.51±0.52 | 0.54±0.44 | 0.12±0.13 | 0.27±0.14 | 0.26±0.19 |
| **BBOA** | 3.09±3.78 | 0.21±0.22 | 0.21±0.18 | 8.32±5.58 | 9.46±4.65 | 6.76±3.19 | 1.35±1.20 | 0.66±0.95 | 0.50±0.54 | 0.14±0.11 | 0.21±0.11 | 0.19±0.08 |
| **SCOA** | 0.61±0.68 | 0.13±0.11 | 0.08±0.03 | 0.48±0.59 | 0.47±0.26 | 0.06±0.06 | 0.23±0.18 | 0.46±0.37 | 0.14±0.09 | 0.17±0.27 | 0.12±0.06 | 0.06±0.04 |
| **PBOA** | 2.04±0.96 | 1.82±0.75 | 0.24±0.14 | 0.66±0.57 | 1.60±0.68 | 1.01±0.72 | 1.02±0.58 | 1.12±0.45 | 0.38±0.25 | 1.63±0.64 | 1.99±0.51 | 0.31±0.22 |
| **FOOA** | 0.79±0.68 | 0.89±0.63 | 0.22±0.19 | 1.35±0.69 | 1.37±0.53 | 0.98±0.27 | 1.38±0.70 | 1.06±1.01 | 0.66±0.66 | 0.70±0.65 | 0.75±0.44 | 0.36±0.27 |
| **SOOA** | 1.16±1.13 | 2.05±0.67 | 2.22±0.78 | 0.29±0.21 | 0.31±0.25 | 0.43±0.38 | 1.05±0.77 | 1.44±0.70 | 1.59±0.76 | 2.41±1.08 | 1.97±0.90 | 2.37±0.82 |
| **WOOA** | 0.64±0.62 | 0.33±0.30 | 0.49±0.47 | 2.02±1.75 | 1.27±1.03 | 0.49±0.49 | 1.98±0.88 | 0.49±0.79 | 0.59±1.17 | 0.59±0.51 | 0.21±0.19 | 0.23±0.22 |



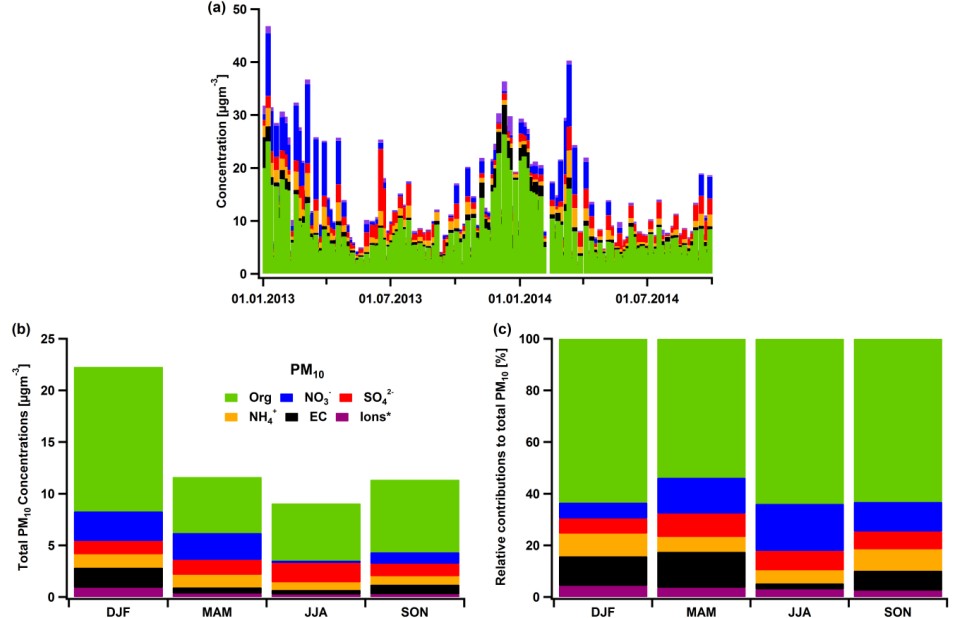

Figure 1. Concentrations of OM, EC and major ionic species for the years 2013, 2014 (a), their seasonal concentrations (b) and relative contributions to the total particulate matter ($PM_{10}$) (c). The sum of the ions $Na^+$, $K^+$, $Mg^{2+}$, $Ca^{2+}$ and $Cl^-$ are included in the indication "Ions*".

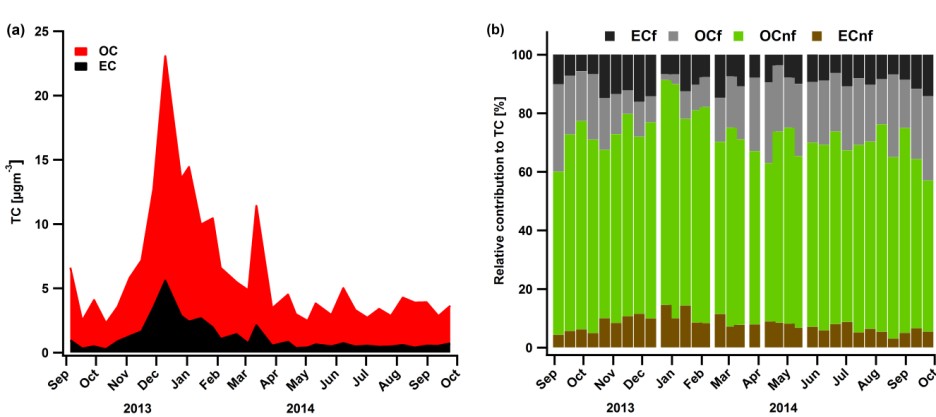

Figure 2. Time series of OC and EC (a) concentrations in $PM_{10}$. $^{14}C$ analysis results with the relative contributions of EC fossil, OC fossil, OC non-fossil and EC non-fossil to the TC (b).





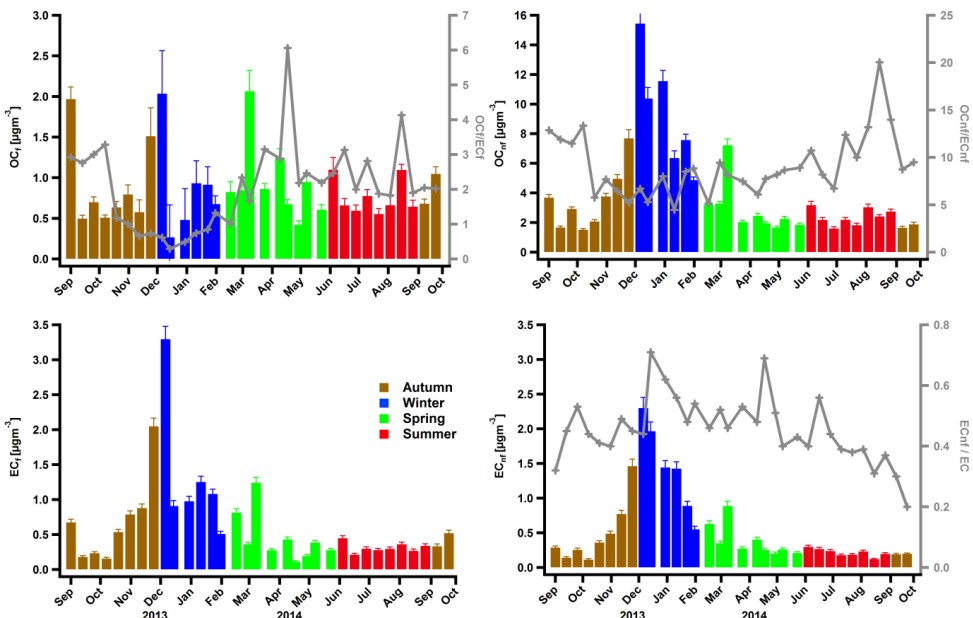

Figure 3. Concentrations in $PM_{10}$ of $OC_f$ (a), $OC_{nf}$ (b), $EC_f$ (c) and $EC_{nf}$ (d) colour-coded by seasons. The ratios $OC_f/EC_f$, $OC_{nf}/EC_{nf}$ and $EC_{nf}/EC$ are also displayed in (a), (b) and (d), respectively.

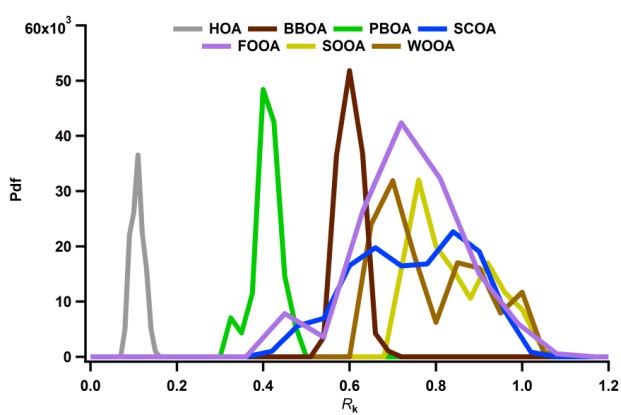

Figure 4. Probability density functions of factor recoveries.





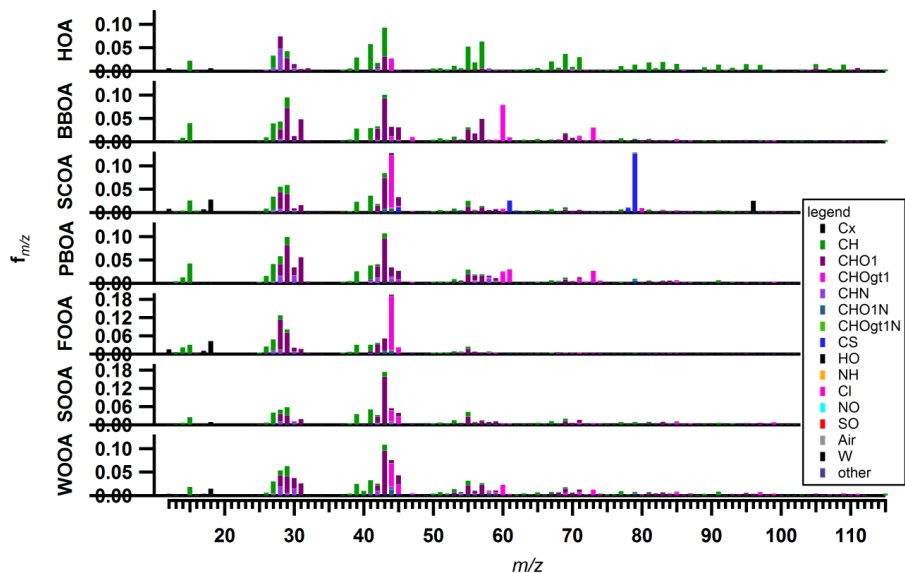

Figure 5. Offline AMS/PMF (ME-2) factor profiles: hydrocarbon like OA (HOA), biomass burning OA (BBOA), sulfur containing OA (SCOA), primary biological OA (PBOA), fossil oxygenated OA (FOOA), summer oxygenated OA (SOOA) and winter oxygenated OA (WOOA).

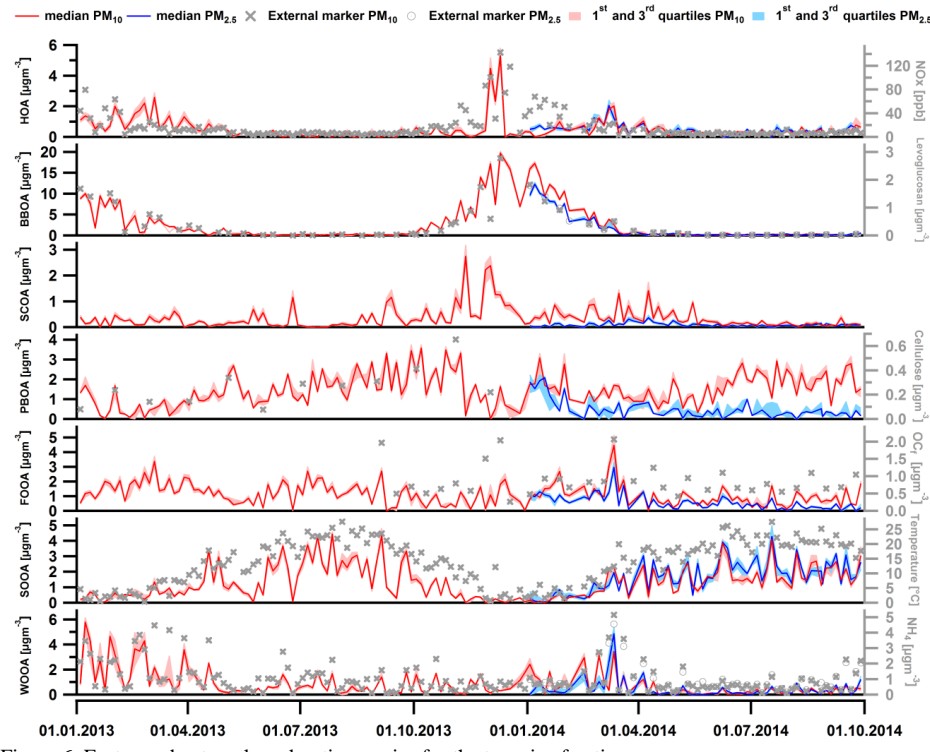

Figure 6. Factor and external marker time-series for the two size fractions.





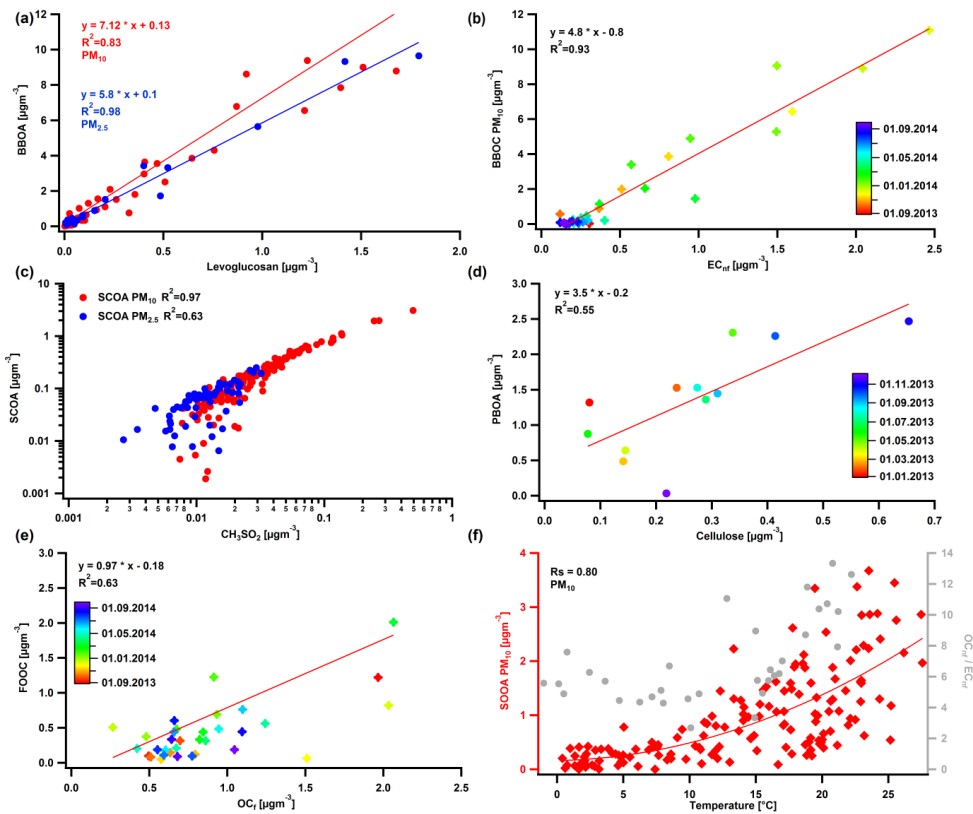

Figure 7. Correlations between BBOA and levoglucosan for the two size fractions (a), BBOC and $EC_{nf}$ for $PM_{10}$ (b), SCOA and $CH_3SO_2^+$ for the two size fractions (c) (the regression line shows a linear relationship), PBOA and cellulose for $PM_{10}$ (d) FOOC and $OC_f$ weighted by the FOOC errors (e) and SOOA and daily averaged temperature as well as $OC_{nf}/EC_{nf}$ ratio and temperature for $PM_{10}$ (f).

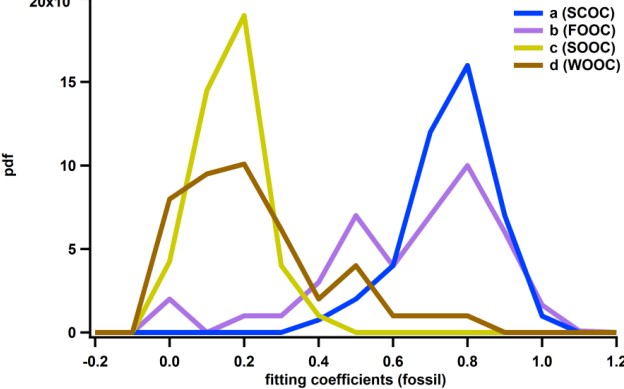

Figure 8. Probability density functions of the fitting coefficients of the relative fossil contributions.





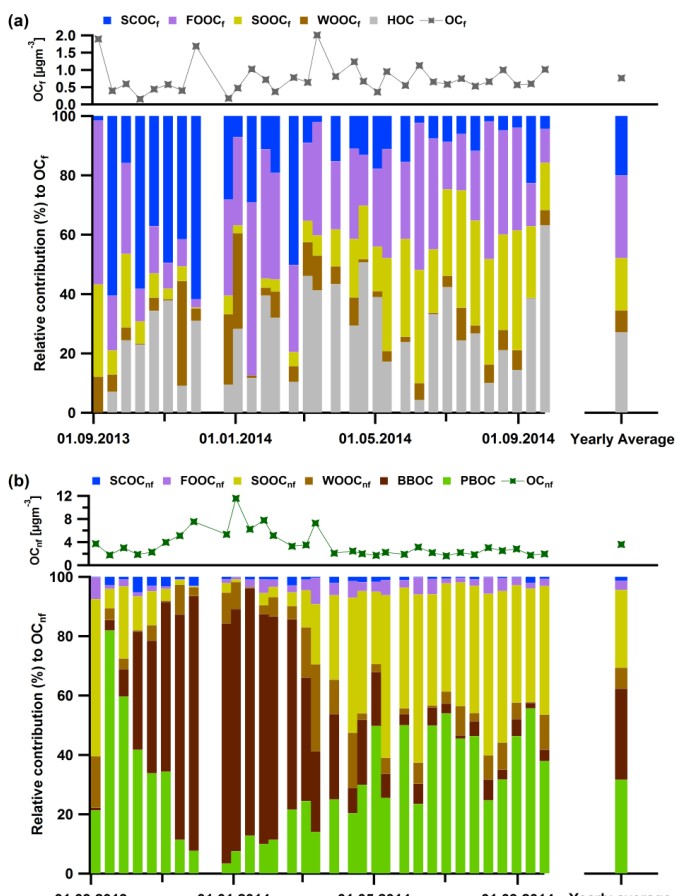

Figure 9. Relative contributions to the fossil OC per factor ($PM_{10}$) (a) and to the non-fossil OC per factor ($PM_{10}$) (b). Note that the total non-fossil concentrations are on average 6 times higher compared to the fossil ones.

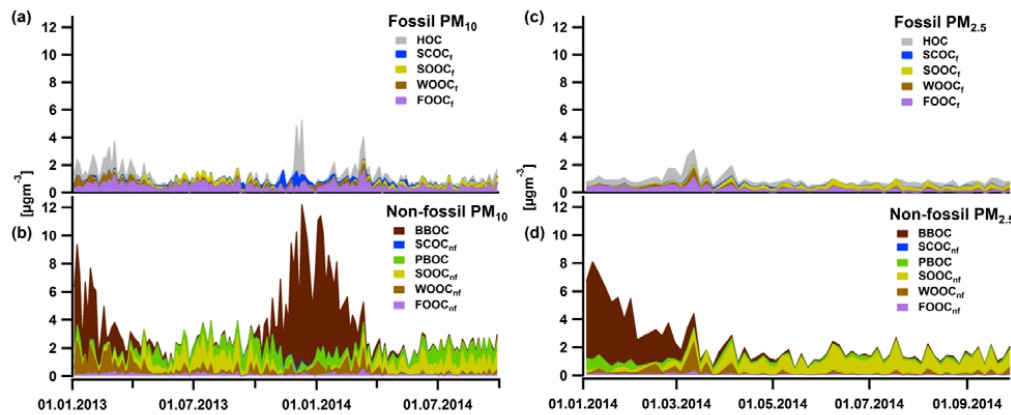

Figure 10. Yearly cycles of fossil $PM_{10}$ (a), non-fossil $PM_{10}$ (b), fossil $PM_{2.5}$ (c), and non-fossil $PM_{2.5}$ (d) OC factors. Note that the covered time periods in (a/b) and (c/d) are different.

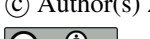


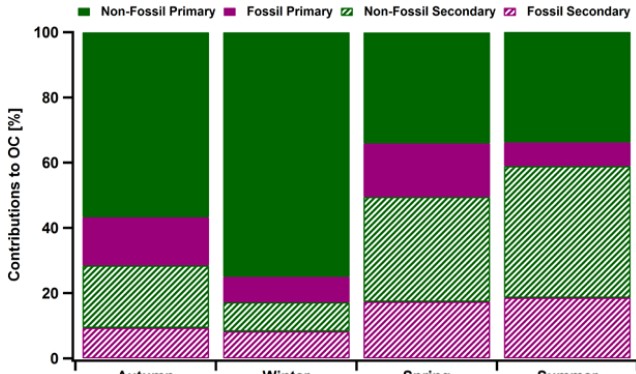

Figure 11. Averaged contributions of the fossil and non-fossil primary and secondary OC to the total OC season-wise for $PM_{10}$.