# Peer review of "Advanced source apportionment of carbonaceous aerosols by coupling offline AMS and radiocarbon size segregated measurements over a nearly two-year period."

_Atmospheric Chemistry and Physics, 2017_

## Referee Comment (RC1) · Anonymous Referee #1 · 12 Jan 2018

The manuscript presents the application of two methodologies representing fundamentally different principles and time resolutions. In a sense, the two distinct methods are complementing each other as one gives information on bulk carbon (a significant part of which is non-soluble) whereas off-line AMS technique represents the water-soluble organic and inorganic fractions. It is a real challenge to combine the results of such distinct methodologies to get valuable insight into major factors determining PM source apportionment at that particular location, but it is done correctly and in a scientifically correct way in the manuscript. The methods including statistical processing of the

results are up-to-date and well-founded, uncertainties are handled properly and the conclusions drawn are self-consistent and in a sense rather trivial and correspond to what can be dictated by common sense. There are, however, two minor issues that leave some degree of discomfort in the referee upon reviewing the manuscript.

The first is that in the Introduction it is explicitly implied by the strongly biased selection of references (Page 2, Line 35) that the whole story of using miniaturised radiocarbon measurements for source apportionment of carbonaceous aerosols has started around 2010 only and been carried out exclusively by groups affiliated to the authors of this manuscript. The fact is that such studies have started around 2000 (see e.g. Lemire et al. JGR 2002), and were also carried out in Europe already at that time (in fact by the group of the authors themselves Szidat et al., 2004) and even within a large scale European project (e.g. Gelencser et al., 2007 JGR). The major conclusions of the latter study were very much in tune with the main findings of this manuscript. Apart from the radiocarbon-based source apportionment studies there have been other studies based on other principles such as specific tracers, OC/EC ratios, inverse modelling and the like, which also pinpoint to the growing contribution of biomass burning to PM aerosols even in highly urbanized areas in Europe. It would be fair to quote some of them in the manuscript, which would also strengthen the conclusions of the manuscript.

The second is that since this study is confined to a single location with specific orography and local meteorology and covers a sufficiently long period of time, it is more than tempting that the major findings of the study be tested against the results of inverse modelling using (local) emission inventories. I understand that such an approach is outside the scope of the present manuscript, but maybe a follow-up paper would make use of the very same data and would yield extremely valuable information for such exercises.

---

## Referee Comment (RC2) · Anonymous Referee #2 · 26 Jan 2018

The manuscript presents results from an analysis of atmospheric filter samples collected during 2013 and 2014 in Switzerland using offline HR-ToF-AMS and carbon-14 measurements. The results give increased insights into the sources and types of aerosols observed. Especially interesting is the focus on the type/source of the precursor for the factors instead of the more commonly used degree of oxidation or volatility. The methods and the descriptions of the data analysis are very thorough and a good deal of work is done in calculating and communicating the uncertainties. This manuscript presents results that follow expected trends in the formation and process-

ing of atmospheric aerosols and thus serves as a good demonstration of the feasibility of combining these two analyses. I recommend addressing two minor issues.

1) A mention of blanks is made with respect to the radiocarbon analysis, but there is no discussion of how blanks were handled for the AMS analysis. Were blanks extracted and prepared in the same manner as AMS samples? How did the authors account for the fact that dilute solutions may not show aerosol signal in the AMS when atomized, despite there being some level of organic material in the solution?

2) The authors could increase readability of the manuscript by providing the names corresponding to acronyms in the text the first time the acronyms are used. This includes the factors as well as all components in equations. Also, the letter labels (a,b,c, and d) are missing on Figure 3. It would also be beneficial to have names for the factors in all of the corresponding figure captions.

---

## Author Comment (AC1) · 23 Mar 2018

**Author's response:**

We thank Referee #1 for the careful revision and comments which helped improving the overall quality of the manuscript. A point-by-point answer (in regular typeset) to the referees' remarks (in the *italic typeset*) follows. Changes to the manuscript are indicated in blue font.

**Anonymous Referee #1**

*The manuscript presents the application of two methodologies representing fundamentally different principles and time resolutions. In a sense, the two distinct methods are complementing each other as one gives information on bulk carbon (a significant part of which is non-soluble) whereas off-line AMS technique represents the water-soluble organic and inorganic fractions. It is a real challenge to combine the results of such distinct methodologies to get valuable insight into major factors determining PM source apportionment at that particular location, but it is done correctly and in a scientifically correct way in the manuscript. The methods including statistical processing of the results are up-to-date and well-founded, uncertainties are handled properly and the conclusions drawn are self-consistent and in a sense rather trivial and correspond to what can be dictated by common sense. There are, however, two minor issues that leave some degree of discomfort in the referee upon reviewing the manuscript.*

> *1) The first is that in the Introduction it is explicitly implied by the strongly biased selection of references (Page 2, Line 35) that the whole story of using miniaturised radiocarbon measurements for source apportionment of carbonaceous aerosols has started around 2010 only and been carried out exclusively by groups affiliated to the authors of this manuscript. The fact is that such studies have started around 2000 (see e.g. Lemire et al. JGR 2002), and were also carried out in Europe already at that time (in fact by the group of the authors themselves Szidat et al., 2004) and even within a large scale European project (e.g. Gelencser et al., 2007 JGR). The major conclusions of the latter study were very much in tune with the main findings of this manuscript. Apart from the radiocarbon-based source apportionment studies there have been other studies based on other principles such as specific tracers, OC/EC ratios, inverse modelling and the like, which also pinpoint to the growing contribution of biomass burning to PM aerosols even in highly urbanized areas in Europe. It would be fair to quote some of them in the manuscript, which would also strengthen the conclusions of the manuscript.*

According to the suggestions of anonymous referee#1 we changed the text in the introduction (Page 2, 4th paragraph) as follows:

The radiocarbon ($^{14}$C) analysis of particulate matter has proven to be a powerful technique providing an unequivocal distinction between non-fossil (e.g. biomass burning and biogenic emissions) and fossil (e.g. traffic exhaust emissions and coal burning) sources (Lemire et al., 2002, Szidat et al., 2004, 2009). The measurement of the $^{14}$C content of total carbon (TC), which comprises the elemental carbon (EC) originating from combustion sources and the organic carbon (OC), had been the subject of many studies(Schichtel et al., 2008, Glasius et al., 2011, Genberg et al., 2011, Zotter et al., 2014b, Zhang et al., 2012, 2016, Bonvalot et al., 2016). Results have shown that in European sites especially in Alpine valleys, the non-fossil sources play an important role during winter due to biomass burning and in summer due to biogenic sources (Gelencsér et al., 2007, Zotter et al., 2014b). Moreover, at regional background sites close to urbanised areas in Europe (Dusek et al., 2017) as well as in megacities like Los Angeles and Beijing fossil OA may also exhibit significant contributions to the total OA (Zotter et al, 2014a, Zhang et al., 2017). However, the determination of the $^{14}$C content in EC and OC separately is challenging and therefore not often attempted for extended datasets.

In Page 8 Line 22 we added two more citations in the text: Genberg et al. (2011) who reported yearly cycles and used in addition levoglucosan measurements and a chemical transport model and Gilardoni et al. (2011) who as well reported yearly cycles and used back trajectories analysis in addition to the radiocarbon and marker analysis.

So far radiocarbon results have been reported mostly for relatively short periods of time (Bonvallot et al., 2016), mainly describing high concentration events and only few studies report measurements on a yearly basis (Genberg et al., 2011, Gilardoni et al., 2011, Zotter et al., 2014b, Zhang et al., 2016; Zhang et al., 2017; Dusek et al., 2017). Here, for a subset of 33 PM$_{10}$ filters from the year 2014, we present yearly contributions of OC$_{nf}$, OC$_f$, EC$_{nf}$ and EC$_f$.

To compare our results of the residential wood burning with other studies that not only used $^{14}$C analysis but other methods as well, we used the following citations in Page 8 Line 36: Jaffrezo et al., 2005 and Favez et al., 2010 and added Puxbaum et al., 2007 and Sandradewi et al., 2008 (for the aethalometer model).

$OC_{nf}$ was the dominant part of TC throughout the year with contributions of up to 80% in winter and 71% in summer (Fig. 2b) and average concentrations of 8.5±4.2 µg m$^{-3}$ and 2.4±0.6 µg m$^{-3}$ in winter and summer, respectively (Fig. 3b). Such high contributions in winter strongly indicate that biomass burning (BB) from residential heating is the main source of carbonaceous aerosols in this region, similar to previous reports (Jaffrezo et al., 2005, Puxbaum et al., 2007, Sandradewi et al., 2008, Favez et al., 2010, Zotter et al., 2014b). The coefficient of determination $R^2$ between $OC_{nf}$ and levoglucosan, a characteristic marker for BB, was 0.92 (Fig. S7a) and the slope ($OC_{nf}$/levoglucosan = 4.8±0.3) lies within the reported range by Zotter et al. (2014b) for Magadino (which was 6.9±2.6).

2) *The second is that since this study is confined to a single location with specific orography and local meteorology and covers a sufficiently long period of time, it is more than tempting that the major findings of the study be tested against the results of inverse modelling using (local) emission inventories. I understand that such an approach is outside the scope of the present manuscript, but maybe a follow-up paper would make use of the very same data and would yield extremely valuable information for such exercises.*

We agree with the reviewer that comparing our results to a modelling study is valuable. However, modelling meteorological parameters over a mountainous region is challenging due to spatial resolution limitations for example, a potential alternation of the type of land within one grid. Moreover, in some cases during winter the planetary boundary layer height ends below the measurement stations and therefore a mismatch between measurements and model often occurs in such regions (Ciarelli et al., 2016, Freney et al., 2011). For these reasons, such comparisons are rarely conducted for Alpine regions and would need the development of highly resolved models for specifically this region.

[revised manuscript text omitted]

---

## Author Comment (AC2) · 23 Mar 2018

**Author's response:**

We thank Referee #2 for the careful revision and comments which helped in improving the overall quality of the manuscript.

A point-by-point answer (in regular typeset) to the referees' remarks (in the *italic typeset*) follows, while changes to the manuscript are indicated in blue font.

**Anonymous Referee #2**

*The manuscript presents results from an analysis of atmospheric filter samples collected during 2013 and 2014 in Switzerland using offline HR-ToF-AMS and carbon 14 measurements. The results give increased insights into the sources and types of aerosols observed. Especially interesting is the focus on the type/source of the precursor for the factors instead of the more commonly used degree of oxidation or volatility. The methods and the descriptions of the data analysis are very thorough and a good deal of work is done in calculating and communicating the uncertainties. This manuscript presents results that follow expected trends in the formation and processing of atmospheric aerosols and thus serves as a good demonstration of the feasibility of combining these two analyses. I recommend addressing two minor issues.*

1) *A mention of blanks is made with respect to the radiocarbon analysis, but there is no discussion of how blanks were handled for the AMS analysis. Were blanks extracted and prepared in the same manner as AMS samples? How did the authors account for the fact that dilute solutions may not show aerosol signal in the AMS when atomized, despite there being some level of organic material in the solution?*

Indeed, in the offline AMS analysis the field blanks were extracted and prepared in the same way as with the samples. In several studies in the past (Bozzetti et al., 2016, 2017a, Daellenbach et al., 2017) field blanks were measured and compared to the nebulized ultrapure water. The resulting signal of the field blank, as in our case, was not statistically different from that of nebulized Milli-Q water.

To ensure that particles generated from dilute solutions are not smaller than the AMS lens transmission and could be measured, we have nebulized $NH_4NO_3$ and $(NH_4)_2SO_4$ solutions (1ppm), providing additional material in the blank. For a number of *m/z* (45%), the resulting signals are statistically significantly higher than nebulized Milli-Q water (by up to a factor of two), but remain negligible compared to ambient filter signals (on average by a factor of 120). As some of this signal can arise from additional operations during solution preparation (e.g. impurities in the salts or different materials (glassware) used for the salt solution preparation compared to the sample preparation) and as the associated signals are negligible (<1% of the signals), we do not correct the filter measurements for the blanks obtained using nebulized $NH_4NO_3$ and $(NH_4)_2SO_4$ solutions.

2) *The authors could increase readability of the manuscript by providing the names corresponding to acronyms in the text the first time the acronyms are used. This includes the factors as well as all components in equations. Also, the letter labels (a, b, c, and d) are missing on Figure 3. It would also be beneficial to have names for the factors in all of the corresponding figure captions.*

Corrected as suggested in:

Page 4 Lines 35 and 36: $\max ATN_{S_3}$ is the maximum attenuation in step three, while $initial\ ATN_{S_2}$ and $initial\ ATN_{S_1}$ are the initial attenuations in step two and one, respectively.

Page 6 Line 11: water soluble organic matter ($WSOM_i$)

Page 7 Line 7: and $\left(\frac{OM}{OC}\right)_{bulk}$ is estimated from the input data matrix for the PMF.

Page 7 Line 11: Where $\left(\frac{OM}{OC}\right)_k$ is calculated from each factor profile.

Page 9 Line 36: Hydrocarbon like OA (HOA)

Page 9 Line 46: Biomass burning OA (BBOA)

Page 10 Line 12: Sulphur containing OA (SCOA)

Page 10 Line 23: Primary biological OA (PBOA)

Page 10 Line 40; note here as well the changed nomenclature: anthropogenic OOA (AOOA).

Page 10 Line 47: Summer oxygenated OA (SOOA)

Page 11 Line 12: Named after its seasonal behavior (Daellenbach et al. 2017), the third oxygenated factor, winter oxygenated OA (WOOA)

Page 12 Line 22: The fossil fractions of SOOC ($SOOC_f$) and WOOC ($WOOC_f$)

Page 12 Line 27: From the non-fossil sources, apart from non-fossil SCOC ($SCOC_{nf}$) and non-fossil OOC ($OOC_{nf}$)

Page 12 Lines 30, 31: SOOC was 79% non-fossil which supported the AMS/PMF results: the significance of non-fossil SOOC ($SOOC_{nf}$)

Page 12 Line 40: Non-fossil WOOC ($WOOC_{nf}$)

Figure 3 was also corrected and names of factors in all corresponding figure captions were added.

Main manuscript, Figure 4 caption: Probability density functions of factor recoveries: hydrocarbon like OA (HOA) in grey, biomass burning OA (BBOA) in dark brown, sulphur containing OA (SCOA) in blue, primary biological OA (PBOA) in green, anthropogenic oxygenated OA (AOOA) in purple, summer oxygenated OA (SOOA) in yellow and winter oxygenated OA (WOOA) in light brown.

Main manuscript, Figure 6 caption: Factor (in red for $PM_{10}$ and blue for $PM_{2.5}$) and external marker (in grey markers) time-series for the two size fractions: HOC and NOx, BBOC and levoglucosan, SCOC, PBOC and cellulose, AOOC and $OC_f$, SOOC and temperature and WOOC and $NH_4^+$.

Main manuscript, Figure 8 caption: Probability density functions of the fitting coefficients of the relative fossil contributions: SCOC in blue, AOOC in purple, SOOC in yellow and WOOC in light brown.

Main manuscript, Figure 9 caption: Relative contributions to the fossil OC per factor ($PM_{10}$) (a) and to the non-fossil OC per factor ($PM_{10}$) (b): BBOC in dark brown, $SCOC_f$ and $SCOC_{nf}$ in blue, PBOC in green, $AOOC_f$ and $AOOC_{nf}$ in purple, $SOOC_f$ and $SOOC_{nf}$ in yellow and $WOOC_f$ and $WOO_{nf}$ in light brown. Note that the total non-fossil concentrations (dark green markers) are on average 6 times higher compared to the fossil ones (dark grey markers).

Main manuscript, Figure 10 caption: Yearly cycles of fossil $PM_{10}$ (a), non-fossil $PM_{10}$ (b), fossil $PM_{2.5}$ (c), and non-fossil $PM_{2.5}$ (d) OC factors: BBOC in dark brown, $SCOC_f$ and $SCOC_{nf}$ in blue, PBOC in green, $AOOC_f$ and $AOOC_{nf}$ in purple, $SOOC_f$ and $SOOC_{nf}$ in yellow and $WOOC_f$ and $WOO_{nf}$ in light brown. Note that the covered time periods in (a/b) and (c/d) are different.

**References**

Bozzetti, C., Daellenbach, K., R., Hueglin, C., Fermo, P., Sciare, J., Kasper-Giebl, A., Mazar, Y., Abbaszade, G., El Kazzi, M., Gonzalez, R., Shuster Meiseles, T., Flasch, M., Wolf, R., Křepelová, A., Canonaco, F., Schnelle-Kreis, J., Slowik, J. G., Zimmermann, R., Rudich, Y., Baltensperger, U., El Haddad, I., and Prévôt, A. S. H.: Size-resolved identification, characterization, and quantification of primary biological organic aerosol at a European rural site, Environ. Sci. Technol., 50, 3425-3434, doi:10.1021/acs.est.5b05960, 2016.

Bozzetti, C., Sosedova, Y., Xiao, M., Daellenbach, K. R., Ulevicius, V., Dudoitis, V., Mordas, G., Byčenkienė, S., Plauškaitė, K., Vlachou, A., Golly, B., Chazeau, B., Besombes, J.-L., Baltensperger, U., Jaffrezo, J.-L., Slowik, J. G., El Haddad, I., and Prévôt, A. S. H.: Argon offline-AMS source apportionment of organic aerosol over yearly cycles for an urban, rural, and marine site in northern Europe, Atmos. Chem. Phys., 17, 117-141, doi:10.5194/acp-17-117-2017, 2017a.

Daellenbach K. R., Stefenelli G., Bozzetti C., Vlachou A., Fermo P., Gonzalez R., Piazzalunga A., Colombi C., Canonaco F., Kasper-Giebl A., Jaffrezo J.-L., Bianchi F., Slowik J. G., Baltensperger U., El-Haddad I., and Prévôt A. S. H.: Long-term chemical analysis and organic aerosol source apportionment at 9 sites in Central Europe: Source identification and uncertainty assessment, Atmos. Chem. Phys., 17, 13265-13282, doi:10.5194/acp-2017-124, 2017.

---

## Author Response (AR1)

We would like to thank the editor and note the following points:

- A point-by-point answer (in regular typeset) to the referees' remarks (in the *italic typeset*) follows. Changes to the manuscript are indicated in blue font.
- Apart from the changes implemented as responses to the referees, after internal discussions we decided to change the name of the factor fossil oxygenated organic aerosol (FOOA) which can be misleading when it comes to the coupling of the offline AMS and radiocarbon techniques. The name of the factor was changed to anthropogenic oxygenated organic aerosol (AOOA). All the changes in the main manuscript and the supplementary are distinguished with the "track changes" tool of Word.
- Some other minor corrections were also implemented and can be again distinguished with the "track changes" tool of Word.

**Author's response:**

We thank Referee #1 for the careful revision and comments which helped improving the overall quality of the manuscript. A point-by-point answer (in regular typeset) to the referees' remarks (in the *italic typeset*) follows. Changes to the manuscript are indicated in blue font.

**Anonymous Referee #1**

*The manuscript presents the application of two methodologies representing fundamentally different principles and time resolutions. In a sense, the two distinct methods are complementing each other as one gives information on bulk carbon (a significant part of which is non-soluble) whereas off-line AMS technique represents the water-soluble organic and inorganic fractions. It is a real challenge to combine the results of such distinct methodologies to get valuable insight into major factors determining PM source apportionment at that particular location, but it is done correctly and in a scientifically correct way in the manuscript. The methods including statistical processing of the results are up-to-date and well-founded, uncertainties are handled properly and the conclusions drawn are self-consistent and in a sense rather trivial and correspond to what can be dictated by common sense. There are, however, two minor issues that leave some degree of discomfort in the referee upon reviewing the manuscript.*

*1) The first is that in the Introduction it is explicitly implied by the strongly biased selection of references (Page 2, Line 35) that the whole story of using miniaturised radiocarbon measurements for source apportionment of carbonaceous aerosols has started around 2010 only and been carried out exclusively by groups affiliated to the authors of this manuscript. The fact is that such studies have started around 2000 (see e.g. Lemire et al. JGR 2002), and were also carried out in Europe already at that time (in fact by the group of the authors themselves Szidat et al., 2004) and even within a large scale European project (e.g. Gelencser et al., 2007 JGR). The major conclusions of the latter study were very much in tune with the main findings of this manuscript. Apart from the radiocarbon-based source apportionment studies there have been other studies based on other principles such as specific tracers, OC/EC ratios, inverse modelling and the like, which also pinpoint to the growing contribution of biomass burning to PM aerosols even in highly urbanized areas in Europe. It would be fair to quote some of them in the manuscript, which would also strengthen the conclusions of the manuscript.*

According to the suggestions of anonymous referee#1 we changed the text in the introduction (Page 2, 4[th] paragraph) as follows:

The radiocarbon ($^{14}$C) analysis of particulate matter has proven to be a powerful technique providing an unequivocal distinction between non-fossil (e.g. biomass burning and biogenic emissions) and fossil (e.g. traffic exhaust emissions and coal burning) sources (Lemire et al., 2002, Szidat et al., 2004, 2009). The measurement of the $^{14}$C content of total carbon (TC), which comprises the elemental carbon (EC) originating from combustion sources and the organic carbon (OC), had been the subject of many studies(Schichtel et al., 2008, Glasius et al., 2011, Genberg et al., 2011, Zotter et al., 2014b, Zhang et al., 2012, 2016, Bonvalot et al., 2016). Results have shown that in European sites especially in Alpine valleys, the non-fossil sources play an important role during winter due to biomass burning and in summer due to biogenic sources (Gelencsér et al., 2007, Zotter et al., 2014b). Moreover, at regional background sites close to urbanised areas in Europe (Dusek et al., 2017) as well as in megacities like Los Angeles and Beijing fossil OA may also exhibit significant contributions to the total

OA (Zotter et al, 2014a, Zhang et al., 2017). However, the determination of the $^{14}$C content in EC and OC separately is challenging and therefore not often attempted for extended datasets.

In Page 8 Line 22 we added two more citations in the text: Genberg et al. (2011) who reported yearly cycles and used in addition levoglucosan measurements and a chemical transport model and Gilardoni et al. (2011) who as well reported yearly cycles and used back trajectories analysis in addition to the radiocarbon and marker analysis.

So far radiocarbon results have been reported mostly for relatively short periods of time (Bonvallot et al., 2016), mainly describing high concentration events and only few studies report measurements on a yearly basis (Genberg et al., 2011, Gilardoni et al., 2011, Zotter et al., 2014b, Zhang et al., 2016; Zhang et al., 2017; Dusek et al., 2017). Here, for a subset of 33 $PM_{10}$ filters from the year 2014, we present yearly contributions of $OC_{nf}$, $OC_f$, $EC_{nf}$ and $EC_f$.

To compare our results of the residential wood burning with other studies that not only used $^{14}$C analysis but other methods as well, we used the following citations in Page 8 Line 36: Jaffrezo et al., 2005 and Favez et al., 2010 and added Puxbaum et al., 2007 and Sandradewi et al., 2008 (for the aethalometer model).

$OC_{nf}$ was the dominant part of TC throughout the year with contributions of up to 80% in winter and 71% in summer (Fig. 2b) and average concentrations of 8.5±4.2 µg m$^{-3}$ and 2.4±0.6 µg m$^{-3}$ in winter and summer, respectively (Fig. 3b). Such high contributions in winter strongly indicate that biomass burning (BB) from residential heating is the main source of carbonaceous aerosols in this region, similar to previous reports (Jaffrezo et al., 2005, Puxbaum et al., 2007, Sandradewi et al., 2008, Favez et al., 2010, Zotter et al., 2014b). The coefficient of determination $R^2$ between $OC_{nf}$ and levoglucosan, a characteristic marker for BB, was 0.92 (Fig. S7a) and the slope ($OC_{nf}$/levoglucosan = 4.8±0.3) lies within the reported range by Zotter et al. (2014b) for Magadino (which was 6.9±2.6).

2) *The second is that since this study is confined to a single location with specific orography and local meteorology and covers a sufficiently long period of time, it is more than tempting that the major findings of the study be tested against the results of inverse modelling using (local) emission inventories. I understand that such an approach is outside the scope of the present manuscript, but maybe a follow-up paper would make use of the very same data and would yield extremely valuable information for such exercises.*

We agree with the reviewer that comparing our results to a modelling study is valuable. However, modelling meteorological parameters over a mountainous region is challenging due to spatial resolution limitations for example, a potential alternation of the type of land within one grid. Moreover, in some cases during winter the planetary boundary layer height ends below the measurement stations and therefore a mismatch between measurements and model often occurs in such regions (Ciarelli et al., 2016, Freney et al., 2011). For these reasons, such comparisons are rarely conducted for Alpine regions and would need the development of highly resolved models for specifically this region.

**Author's response:**

We thank Referee #2 for the careful revision and comments which helped in improving the overall quality of the manuscript.

A point-by-point answer (in regular typeset) to the referees' remarks (in the *italic typeset*) follows, while changes to the manuscript are indicated in blue font.

**Anonymous Referee #2**

*The manuscript presents results from an analysis of atmospheric filter samples collected during 2013 and 2014 in Switzerland using offline HR-ToF-AMS and carbon 14 measurements. The results give increased insights into the sources and types of aerosols observed. Especially interesting is the focus on the type/source of the precursor for the factors instead of the more commonly used degree of oxidation or volatility. The methods and the descriptions of the data analysis are very thorough and a good deal of work is done in calculating and*

*communicating the uncertainties. This manuscript presents results that follow expected trends in the formation and processing of atmospheric aerosols and thus serves as a good demonstration of the feasibility of combining these two analyses. I recommend addressing two minor issues.*

> *1) A mention of blanks is made with respect to the radiocarbon analysis, but there is no discussion of how blanks were handled for the AMS analysis. Were blanks extracted and prepared in the same manner as AMS samples? How did the authors account for the fact that dilute solutions may not show aerosol signal in the AMS when atomized, despite there being some level of organic material in the solution?*

Indeed, in the offline AMS analysis the field blanks were extracted and prepared in the same way as with the samples. In several studies in the past (Bozzetti et al., 2016, 2017a, Daellenbach et al., 2017) field blanks were measured and compared to the nebulized ultrapure water. The resulting signal of the field blank, as in our case, was not statistically different from that of nebulized Milli-Q water.

To ensure that particles generated from dilute solutions are not smaller than the AMS lens transmission and could be measured, we have nebulized $NH_4NO_3$ and $(NH_4)_2SO_4$ solutions (1ppm), providing additional material in the blank. For a number of $m/z$ (45%), the resulting signals are statistically significantly higher than nebulized Milli-Q water (by up to a factor of two), but remain negligible compared to ambient filter signals (on average by a factor of 120). As some of this signal can arise from additional operations during solution preparation (e.g. impurities in the salts or different materials (glassware) used for the salt solution preparation compared to the sample preparation) and as the associated signals are negligible (<1% of the signals), we do not correct the filter measurements for the blanks obtained using nebulized $NH_4NO_3$ and $(NH_4)_2SO_4$ solutions.

> *2) The authors could increase readability of the manuscript by providing the names corresponding to acronyms in the text the first time the acronyms are used. This includes the factors as well as all components in equations. Also, the letter labels (a ,b, c, and d) are missing on Figure 3. It would also be beneficial to have names for the factors in all of the corresponding figure captions.*

Corrected as suggested in:

Page 4 Lines 35 and 36: max $ATN_{S_3}$ is the maximum attenuation in step three, while $initial\ ATN_{S_2}$ and $initial\ ATN_{S_1}$ are the initial attenuations in step two and one, respectively.

Page 6 Line 11: water soluble organic matter ($WSOM_i$)

Page 7 Line 7: and $\left(\frac{OM}{OC}\right)_{bulk}$ is estimated from the input data matrix for the PMF.

Page 7 Line 11: Where $\left(\frac{OM}{OC}\right)_k$ is calculated from each factor profile.

Page 9 Line 36: Hydrocarbon like OA (HOA)

Page 9 Line 46: Biomass burning OA (BBOA)

Page 10 Line 12: Sulphur containing OA (SCOA)

Page 10 Line 23: Primary biological OA (PBOA)

Page 10 Line 40; note here as well the changed nomenclature: anthropogenic OOA (AOOA).

Page 10 Line 47: Summer oxygenated OA (SOOA)

Page 11 Line 12: Named after its seasonal behavior (Daellenbach et al. 2017), the third oxygenated factor, winter oxygenated OA (WOOA)

Page 12 Line 22: The fossil fractions of SOOC (SOOC$_f$) and WOOC (WOOC$_f$)

Page 12 Line 27: From the non-fossil sources, apart from non-fossil SCOC (SCOC$_{nf}$) and non-fossil OOC (OOC$_{nf}$)

Page 12 Lines 30, 31: SOOC was 79% non-fossil which supported the AMS/PMF results: the significance of non-fossil SOOC (SOOC$_{nf}$)

Page 12 Line 40: Non-fossil WOOC (WOOC$_{nf}$)

Figure 3 was also corrected and names of factors in all corresponding figure captions were added.

[revised manuscript text omitted]

Yet, the PBOA factor was clearly identified with the introduction of a seventh factor which exhibited a distinct enhancement in the coarse fraction in spring and summer. This seven-factor solution resulted in a further decrease of the residuals and the resolution of three oxygenated factors. Two of which were named after their seasonal behaviour, winter and summer OOA (WOOA and SOOA), as proposed by Daellenbach et al. (2017). The third OOA exhibited a rather stable yearly cycle and high contribution at $m/z$ 44 ($CO_2^+$); see below in Fig. S5. As this factor was mainly fossil and correlated with fossil OC (as explained in Section 4.3 of the main text), we called this factor anthropogenic OOA (AOOA). However, note that AOOA was not the only anthropogenic OOA factor; WOOA was also related to non-fossil anthropogenic activities such as wood burning (see Section 4 of main text). Higher order solutions resulted in a further splitting of the oxygenated factors WOOA and AOOA, which could not be interpreted. Hence, we selected this seven-factor solution.

[Figure]

Figure S2. Scatter plot with p-values from the comparison of the two size fractions for the selected solutions of the water soluble factors SOOC and HOC (WSSOOC and WSHOC) (see main text Section 3.3 for conversion of WSOA to WSOC). Some data points from the factors WSSOOC and WSHOC exhibited higher concentrations for the $PM_{2.5}$ size fraction compared to $PM_{10}$, which is not physically possible as aerosols collected by a $PM_{10}$ inlet include also the $PM_{2.5}$ size fraction.

Table S1. P-value range resulting from the correlation between a water soluble factor in $PM_{10}$ and the respective one in $PM_{2.5}$.

| P value range | WSHOA | WSBBOA | WSPBOA | WSSCOA | WSAOOA | WSSOOA | WSWOOA |
|---|---|---|---|---|---|---|---|
| Min | 1.08E-05 | 4.00E-04 | 1.04E-06 | 1.30E-13 | 6.86E-10 | 8.02E-08 | 7.66E-3 |
| Max | 0.99 | 0.33 | 0.01 | 0.77 | 0.33 | 0.94 | 0.99 |

[Figure]

Figure S3. Example of a scatter plot between each bootstrap solution i and their average for the water soluble PBOC (WSPBOC) factor.

**S.2 Recoveries weighting factor**

5    To select the physically meaningful recoveries we applied a weighting factor $f$ calculated by the following equation:

$$f = \begin{cases} 1 & , for\ 0 < R_{i,k,max} \leq 1 \\ \dfrac{\frac{1}{\sigma\sqrt{2\pi}}exp*\left(-\frac{\left(R_{i,k,max}-\mu\right)^2}{2\sigma^2}\right)}{\frac{1}{\sigma\sqrt{2\pi}}exp*\left(-\frac{(1-\mu)^2}{2\sigma^2}\right)} & , for\ R_{i,k,max} > 1 \end{cases} \qquad (S1)$$

Where $\sigma = 0.05$, $\mu = 1$, $i$ the number of iterations and $k$ the factor.

A visualisation of the weighting factor is shown in Figure S4.

[Figure]

Figure S4. Probability of $R_k$ occurrence: all $R_k$ that exhibited values between [0,1] are weighted by 1 and the $R_k >$ 1 are downweighted. Even though $R_k > 1$ is physically not plausible errors in OC and WSOC may allow such values.

[revised manuscript text omitted]